# Upcycling Compact Discs for Flexible and Stretchable Bioelectronic Applications

Matthew S. Brown[1], Louis Somma[1], Melissa Mendoza [1], Yeonsik Noh[2], Gretchen J. Mahler[1] & Ahyeon Koh [1✉]

Electronic waste is a global issue brought about by the short lifespan of electronics. Viable methods to relieve the inundated disposal system by repurposing the enormous amount of electronic waste remain elusive. Inspired by the need for sustainable solutions, this study resulted in a multifaceted approach to upcycling compact discs. The once-ubiquitous plates can be transformed into stretchable and flexible biosensors. Our experiments and advanced prototypes show that effective, innovative biosensors can be developed at a low-cost. An affordable craft-based mechanical cutter allows pre-determined patterns to be scored on the recycled metal, an essential first step for producing stretchable, wearable electronics. The active metal harvested from the compact discs was inert, cytocompatible, and capable of vital biopotential measurements. Additional studies examined the material's resistive emittance, temperature sensing, real-time metabolite monitoring performance, and moisture-triggered transience. This sustainable approach for upcycling electronic waste provides an advantageous research-based waste stream that does not require cutting-edge microfabrication facilities, expensive materials, and high-caliber engineering skills.

[1] Department of Biomedical Engineering, State University of New York at Binghamton, Binghamton, NY 13902, USA. [2] College of Nursing and Department of Electrical and Computer Engineering, University of Massachusetts, Amherst, MA 01003, USA. ✉email: akoh@binghamton.edu

The disposal of electronic waste (e-waste) has become a concerning and growing waste stream driven by the short life cycle of electronics. In 2015, the United Nations established a blueprint for Sustainable Development Goals (SDGs)[1]. The 12th SDG, "Responsible Consumption and Production", seeks to address e-waste challenges by ensuring countries adopt a more responsible approach to the proliferating e-waste stream[2]. Inefficient recycling processes are a global concern for e-waste management as they contribute to an increase in landfill waste and produce toxic pollution[3]. Additionally, Stephan Sicars (Director of the Department of Environment UN Industrial Development Organization) described e-waste as "a serious threat to the environment and human health worldwide"[4]. In 2019, the United Nations documented 1.7 kg per capita of e-waste recycled out of 7.3 kg per capita generated. To ensure the recycling of all e-waste by 2030 the recycling rate will need to be roughly ten times greater[2]. To reduce landfill and pollution accumulation, a more sustainable method is required to manage the flow of e-waste. Currently, only ~15–20% of e-waste is recycled despite its valuable materials—iron, steel, copper, silver, and gold[5–7]. Whereas the remaining 80% of e-waste is not collected for recycling due to expense and lack of a global infrastructure[5–8]. Meanwhile, the toxic and hazardous components of e-waste—mercury, lead, and synthetic resins—threaten the environment and are left to degrade in landfills or incinerated[5–7]. Today, e-waste primarily consists of dated technologies which accounts for the ever-growing trail[5]. Products from years past such as compact disks (CDs), old televisions, and computer monitors are the biggest contributors to e-waste[5]. Since 1999, 9.02 billion CDs have shipped in the United States[9]. In 2021, CD sales increased from the previous year by 1.1% to 40.6 million[10]. However, these statistics do not consider global shipments and only account for music CDs excluding other types such as DVDs, software disks, and video games. Furthermore, the biomedical field utilizes CDs as a primary medium for medical images between both patients and providers. Thus, the global number of CDs produced and circulating globally are expected to be much larger and the end of the CD waste stream remains unclear. As societal dematerialization increases and we shift further towards electronic platforms, where will all these CDs be deposited? The life cycle and disposal of CDs is particularly concerning as they can depolymerize from polycarbonate into their toxic monomer, Bisphenol A (BPA)[11]. Over time, the steady release of BPAs, a possible xenoestrogen, may have negative health and environmental consequences[11,12]. As such, the exploration of e-waste source recycling and upcycling is imperative.

Biointegrated electronics present novel methods for real-time monitoring of pathophysiological progression, health status, and athletic performance through a wide range of biomarkers[13–20]. Translating rigid electronics into soft mechanics for the seamless integration with soft biological tissue can be achieved with thin polymeric substrates (e.g., polyimide and polydimethylsiloxane)[21–23]. By addressing a mechanical mismatch, conventional, rigid metal materials can be transformed into stretchable components by patterning deterministic architectures (e.g., serpentine, wavy, etc.). This augmentation enables deformation and lowers contact impedance by improving the conformability that exists at the interface between electronics and biological tissue such as skin[24,25]. Existing microfabrication techniques for fabricating stretchable, active components have primarily relied on costly and time-consuming printing or lithography-based technologies[14]. Evaporated gold, used for microfabrication and thin-film production, costs an estimated $95 per gram (~125 nm thick films). Processing costs significantly vary by facility, costing between $2,702–$7,298 per use and $59,016–$139,542 annually[26]. The lead time can range from a few

hours to days depending on the complexity of the device. Moreover, these processes require an abundance of volatile compounds (e.g., chemical etchant, photoresist, developer, etc.) that present environmental hazards[27]. Although advanced techniques are superior in many regards, they may not be suitable for rapid prototyping, experimental testing, or one-time-use sensor development, especially in settings with limited instrumentation[28,29]. One-time-use, disposable sensors are in a growing demand for reliable, accessible, and fast measurements, and that can be used anywhere or any time without recalibration or the worry of contamination[28]. This is especially the case in medical diagnostics that have a wide range of applications in point-of-care sensors deemed to replace central laboratories in resource-limited or time-sensitive measurement settings[14,28]. Additionally, there is a need to reduce the complexity and cost of fabricating stretchable electronic prototype devices, which will advance the potential of manufacturing and reduce the required skill level to fabricate[13,29–32]. To date, researchers have explored alternative uses for CDs to develop gold and silver electrodes[33–36], detect metal ions (e.g., Pb, Hg, Cu, etc.)[37–39], screen organic compounds (e.g., DNA, cysteine, dopamine, etc.)[40–42] and quantify oxidizing agents (e.g., hydrogen peroxide, Cl, iodine, etc.)[43–47]. However, the techniques reported thus far fail to demonstrate an application pathway for biosensor platforms and lack the mechanical durability to be practical for wearable applications.

Here, we developed sustainable engineering approaches to upcycle CDs into stretchable and transient electronics that offer an inexpensive, eco-friendly, and rapid fabrication alternative to conventional microfabrication techniques. The development of these biosensors focused on patterning deterministic and stretchable patterns with an affordable craft mechanical cutter. This study presents the translation of CDs into biopotential, electrochemical, resistive, and biodegradable wearable sensors. We propose a fully integrated electrocardiogram (ECG) sensor with patterned CD electrodes that can communicate with a smartphone via Bluetooth. The upcycled soft bioelectronics exhibited biocompatibility with human keratinocytes, demonstrating their safety and successful application with on-skin, biointegrated electronics.

## Results

**Upcycling CDs into stretchable bioelectronics.** A schematic of the upcycling process is presented in Fig. 1a. The mechanical cutter can define metal and polymeric layers with ease and precision down to feature sizes of 25 μm capable of up to 20% strain (Supplementary Figs. 1, 2). In addition to the mechanical cutter, patterning through photolithography and laser engraving was explored (Supplementary Fig. 3). The ease of use, affordability, and rapid development capabilities of the mechanical cutter proved to be the simplest upcycling process. The entire fabrication was completed within 20–30 min without releasing toxic chemicals or needing expensive equipment, costing ~$1.50 per device. The inputs and outputs of the upcycling fabrication process vs. microfabrication are illustrated in Supplementary Fig. 4. The CD was soaked in 40 mL of acetone for 1.5 min, releasing the metal layer by breaking down the polycarbonate substrate (Fig. 1a1 and Supplementary Figs. 5, 6a)[48,49]. However, the concentration within the acetone was undetectable (Supplementary Fig. 6b). The metal from the CD was easily harvested with polyimide (PI) tape, which also serves as the substrate layer in the new device integration to improve the mechanical durability and robustness of the thin metal film (Fig. 1a2). The PI-metal layer was transferred to tattoo paper to serve as a durable but temporary substrate through the patterning process (Fig. 1a3). The tattoo paper-PI-metal was adhered to the cutting mat and patterned with the mechanical cutter as shown in

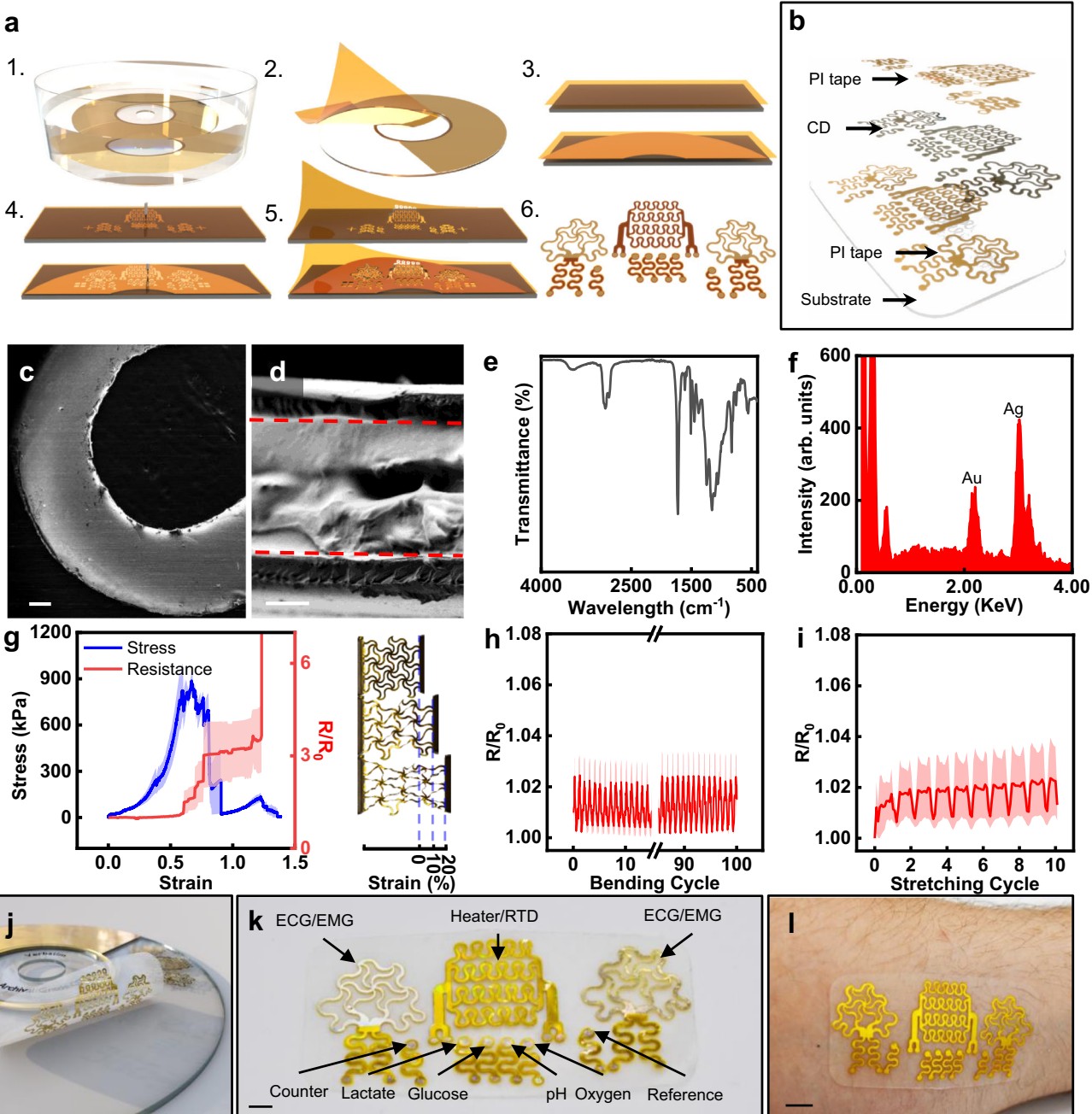

**Fig. 1 Upcycling CDs into stretchable electronics. a** Schematic of the upcycling process. (1) soak in acetone; (2) harvest metal layer with PI tape; (3) (bottom) laminate CD on tattoo paper; (top) PI tape laminated on water-soluble tape; (4) pattern with mechanical cutter; (5) remove excess and laminate insulation layer; (6) UCDEs. **b** Cross-sectional view of the UCDE. SEM image of **c** the CD metal layer after patterning (scale bar, 200 μm) and **d** cross-section of the UCDE (PI-metal-PI)(scale bar, 20 μm). **e** FTIR of the metal layer (PMMA side) after processing in acetone, HCl, and $HNO_3$. **f** Surface characterization with EDS analysis of the CD (metal layer) after soaking in acetone. **g** Mechanical properties as a function of electrical performance, average and standard error of means ($n = 3$) of stress vs. strain (blue) and resistance vs. strain (red). Images of lattice patterned UCDE during tensile testing. Toe region (15 mm length), heel region (19 mm length, 27% strain), and linear region (22.5 mm length, 50% strain). Electrical performance, average and standard error of means ($n = 3$) resistance properties as a function of **h** cyclic bending and **i** cyclic stretching. **j** Representative image of UCDE device. **k** Components of the UCDE device (scale bar, 4 mm). **l** Image of the UCDE device laminated on the skin (scale bar, 1 cm).

Fig. 1a4. Patterns are easily loaded onto the Cricut Design Space software by importing AutoCAD drawings, and the PI-metal layer is carved by the cutting machine (Cricut Maker, USA). Subsequently, insulation layers can be patterned through a similar process by adhering the PI tape to water-soluble tape (Fig. 1a3–4). After processing, the excess material from the metal and insulation layers was removed (Fig. 1a5). The alignment marks allowed the insulation layer to be aligned and laminated onto the metal

layer, thus yielding the upcycled CD electronics (UCDEs) (Fig. 1a6, b). Images of the full process are shown in Supplementary Fig. 7. After processing, the UCDEs demonstrated a base, four-probe resistance of around 0.03 Ω/cm².

The mechanical cutter produced very precise cuts, and the PI tape fully insulates the metal layer (Fig. 1c and Supplementary Fig. 8a, b). The overall thickness of the harvested metal layer from the CD was 30.35 ± 1.92 μm, consisting of a protective, polymethylmethacrylate

(PMMA), and an archival metal layer (~70 nm) (Supplementary Figs. 5, 8c). The PI-metal layer (54.04 ± 2.72 μm) thickness increased with the final insulation layer to 82.24 ± 1.71 μm (Fig. 1d and Supplementary Figs. 8c, d, 9). As presented in Fig. 1e, Fourier-transform infrared spectroscopy (FTIR), revealed that the protective PMMA layer remained intact on the metal layer after the acetone soak, and the layer does not have to be removed to produce the UCDEs and enhanced the durability of the thin archival metal layer (Supplementary Fig. 5). In the FTIR spectrum, the characteristic peaks of PMMA could be identified at 1726 cm$^{-1}$ because of the C=O stretching of the ester group. Bands at 2873 and 2932 cm$^{-1}$ are caused by the C–H stretching of alkanes. Stretching of the C–O–C group was seen at 1060 and 1246 cm$^{-1}$. The weak band at 3468 cm$^{-1}$, attributed to –OH hydroxyl group stretching and bending, is suspected to be physisorbed moisture from the acetone soaking and subsequent DI water washes. Energy-dispersive X-ray spectroscopy (EDS) analysis of the metal layer after the solvent treatments are shown in Fig. 1f and Supplementary Fig. 10. After the soaking in acetone, Ag, and Au could be seen within the spectrum at 70.95 and 29.05 wt%, respectively (Supplementary Fig. 10a, b). Their presence confirmed the archival composition of the layer as predominantly Ag. Additional methods to treat the CD are discussed in the Supporting Information. The CD metal layer can be stripped down to nearly pure gold by soaking in a bath of nitric acid.

Figure 1g–i present the mechanical properties of the UCDEs once they're patterned. Compared to the unpatterned CD, once stretchable features were carved into the device, hyperelastic behavior could be achieved with pertinent deformation strain to human skin (>20% strain)[50] (Fig. 1g; Supplementary Figs. 11, 12; and Supplementary Table 1). The triangular lattice structures ($n = 3$) achieved an elastic modulus and elongation at a yield of 5.59 ± 0.16 MPa and 62.35 ± 1.81%, respectively (Fig. 1g). The elastic modulus of the stress and strain curves of the UCDEs exhibited slightly stiffer mechanics than human skin, E = 10–500 kPa[15] but remain soft enough to be used as stretchable electronics. Furthermore, stretchable patterning enabled cyclic bending and stretching with negligible deviations in resistance (Fig. 1h). Cyclic bending for 100 cycles produced a 0.29% increase in resistance when bent with a bending radius of 3.5 mm. Unpatterned samples presented a larger change in performance with a 21.7% increase in resistance when bent for 100 cycles at a bending radius of 3.5 mm (Supplementary Fig. 12b). Cyclic stretching of the patterned UCDEs for ten cycles induced a 0.59% increase in resistance at a range from 0–20% strain (Fig. 1i).

Because of the strong yield strength and increased durability of the PI tape, the sensors can be laminated onto the skin, substrate-free via liquid bandage. Additionally, the fabricated electrodes can be integrated with a silicone elastomer polymer such as polydimethylsiloxane (PDMS), EcoFlex, or a silicone-based bandage (Fig. 1j–l). The UCDEs can merge with silicone bandages via a hydrolysis-condensation reaction of siloxane to produce a covalent bond. The PI side of the UCDEs can be coated with SiO$_2$ spray (countertop spray sealant) and the hydrolysis-condensation reaction produced by UV ozone treatment bonds the UCDEs to a silicone bandage. A fully fabricated UCDE device consisted of two biopotential electrodes, a heater or temperature sensor, a reference electrode, a counter electrode, a pH electrode, an oxygen electrode, a lactate electrode, and a glucose electrode (Fig. 1k). The full, end-to-end fabrication and manufacturing required resources that can be found easily at conventional craft stores, negating the need for high-end instrumentation.

**Upcycled CD electronics as biopotential sensors**. Figure 2 presents the application of the UCDEs as biopotential sensors.

Biopotential sensors have a wide application for potential use as risk assessments, physical interventions, and diagnostic tools for the brain, heart, or muscle-related diseases through a human-machine interface[51,52]. To demonstrate the performance of the fabricated UCDEs, the recorded biopotential signal was compared to that from commercial gel electrodes. The UCDEs were laminated to the forearm with a liquid bandage for electromyography (EMG) measurements. The commercial gel electrodes were placed directly adjacent to the UCDEs. Two-channel EMG was synchronously recorded using a Quad Bio Amp (PowerLab) with a sampling rate of 1 kHz. The EMG signals presented in Fig. 2a indicate that the two electrode types had similar signal output. The EMG signal captured by the UCDEs had a slightly higher amplitude and could pick up additional motor unit activity. However, we suspect this may be the result of the larger surface area covered by the EMG UCDEs. For ECG measurements, the UCDEs were laminated to the skin with the same technique, however, the electrodes were placed on the left side of the chest at 6 cm apart. The gel electrodes were placed adjacent to the UCDEs on the chest, 6 cm apart. A gel electrode was used as the central ground placed in the upper left abdominal quadrant. Three-channel ECG was recorded with a Quad Bio Amp (PowerLab) at a sampling rate of 1 kHz. The two electrode types presented similar results; however, with the UCDEs, the P and T waves were easier to identify (Fig. 2b). Equivalent to the EMG signal results, we suspect that the larger surface area presented by the UCDEs produced greater signal coverage of the electrical signal from the heart. Figure 2c, d shows the UCDEs as a fully integrated device demonstrated in a fully wireless, wearable, ECG configuration. In this application, the wireless controller was powered by a lithium-polymer battery, consisting of a microcontroller unit (MCU) and Bluetooth module, laminated on top of a silicone bandage. The wireless controller was connected to the UCDEs (laminated on the left chest) and the smartphone application recorded the ECG signal via Bluetooth (Fig. 2e and Supplementary Fig. 13). The signal recorded from the fully functional wireless device presented similar results to the PowerLab system and the characteristic PQRST waves in the ECG signal can all be identified.

**Upcycled CD electronics as an RTD and heater**. We demonstrated the feasibility of the UCDEs as fully stretchable, electrically driven resistive temperature sensors and heaters that have a broad range of applications in healthcare-based settings such as skin temperature sensors[53], blood flow monitors[54], etc[55,56] (Fig. 3). The Joule heating characteristics of the UCDEs as heaters are demonstrated in Fig. 3a–c. Fixed DC bias voltage was applied between the electrode terminals with an incremental increase in applied voltage, 1 V per 30 s (1–7 V). The temperature generated by the UCDE heater, captured by an IR camera, with respect to time at various applied voltages is presented in Fig. 3a. The maximum and average temperatures showed a smooth and responsive output. The maximum temperature was generally concentrated on the end of the device where the resistance was the lowest; however, within the serpentine structure, the heating distribution profile was homogeneous. For wearable applications, we compared our UCDEs to disposable hand warmers, which were determined to emit temperatures up to 42.0 °C with an average of 33.7 °C (Supplementary Fig. 14). Based on the experimental data, a 5 V bias voltage applied to the UCDE heater presented a similar temperature emission to that of commercially available hand warmers (Hot Hands). At 5 V, the heater produced an average heat output of 35.6 °C and a maximum temperature of 52.3 °C. Here, the characteristics of a 2.5 cm wide heater were demonstrated; however, larger sizes could be produced within the dimensions of the CD. A heater 40% larger, performed similarly in temperature output, however, a higher bias

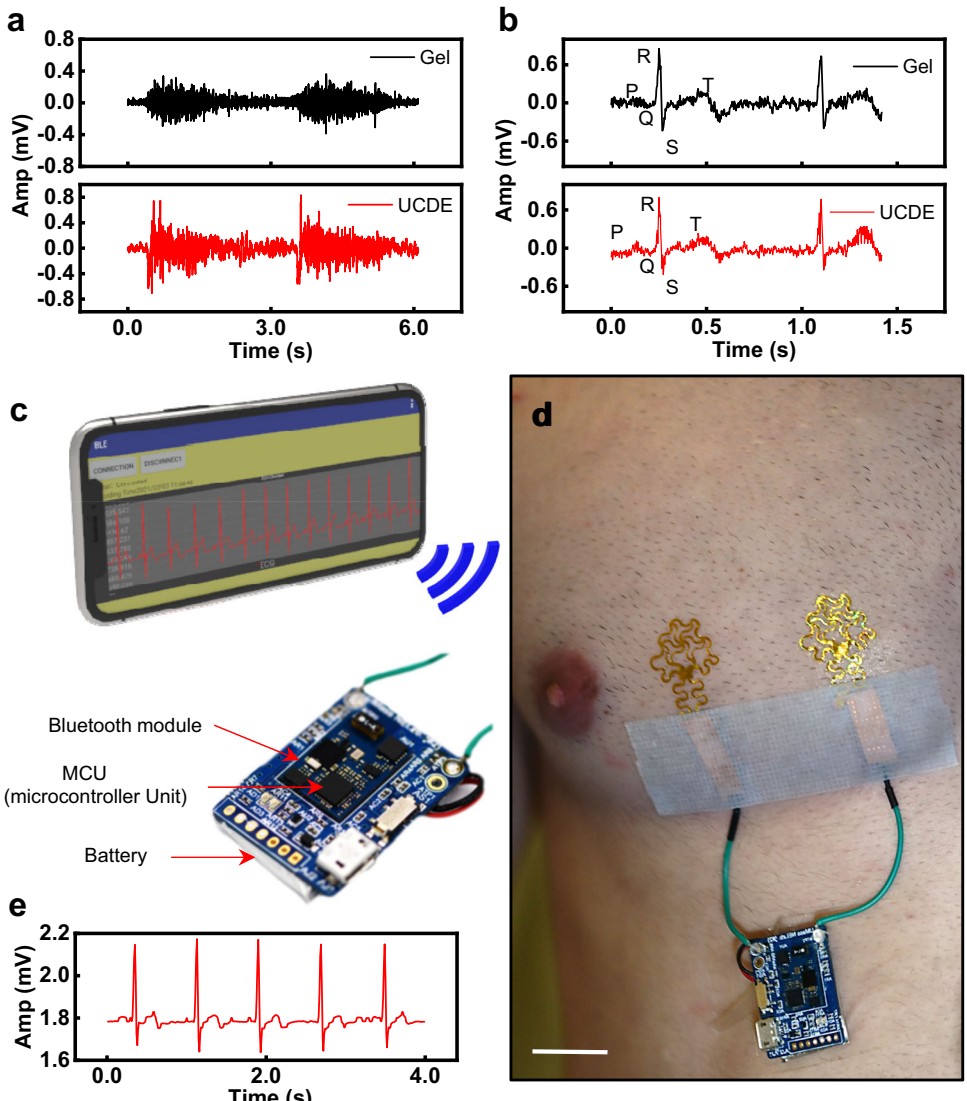

**Fig. 2 Application of UCDEs as stretchable biopotential sensors. a** EMG signal of commercial gel electrodes compared to UCDEs. **b** ECG signal of commercial gel electrodes compared to UCDEs. **c–e** Fully wireless ECG sensor. **c** Schematic illustration of wearable ECG device; a smartphone wirelessly connected via Bluetooth (top) to a controller unit (bottom). **d** Photograph of the stretchable UCDEs as ECG sensors integrated with the wireless operating system (scale bar, 4 cm). **e** ECG signal is recorded via the smartphone application.

voltage was required to achieve a similar temperature profile because of the larger resistance (Supplementary Fig. 15).

Stretchability quantification of the UCDEs heater was explored by examining temperature deviations as a response to tensile strain. At 5 V DC bias voltage, the temperature output was recorded at 0, 10, and 20% strain (Fig. 3b). At a 10% strain, the heater remained within the temperature range of the hand warmer (Hot Hands). From 0% strain to 20% strain, the UCDE suffers from a 19% decrease in average temperature from 35.6 to 28 °C because of the increased resistance across the device. To overcome this performance loss, a stronger voltage could be applied. To achieve a temperature above the 33.7 °C hand warmer, at 20% strain, a 7 V DC could be applied, assuming a 19% loss, and the average output would be 37.2 °C. This could be advantageous for on-skin applications as many regions of the body can produce strains up to 20%[50]. The performance of the UCDE's heater was evaluated by laminating the wearable heater on a subject's palm (Fig. 3c). The UCDE's heater was subjected to extension and flexion at the palm with a 5 V DC voltage applied. When the palm was relaxed, extended, and flexed; the thermal

output of the UCDE's heater performed similarly to in vitro characterization at 5 V DC, with a little, if any change in emitted temperature.

The temperature sensing of the UCDEs was determined by four-probe resistance measurements and calibrated with a thermocouple to develop a resistive temperature detector (RTD) sensor (Fig. 3d, e). The calibration curve of the UCDEs as a temperature sensor is presented in Fig. 3D, change in resistance with respect to change in temperature, a temperature coefficient of $9.21 \times 10^{-4}\,°C^{-1}$ at 20 °C and $R^2$ of 0.99. The temperature sensing response of the UCDEs RTD sensor was evaluated and compared to an IR camera (Fig. 3e). The UCDEs performed analogously to the IR camera with no significant deviations in temperature response time or temperature detection.

**Upcycled CD electronics as electrochemical sensors**. The UCDEs can be evolved into stretchable, electrochemical sensors, with functionality in potentiometric, amperometric, and enzymatic-based biosensors. Figure 4 highlights the electrochemical characteristics of the stretchable UCDE sensors. The electrochemical

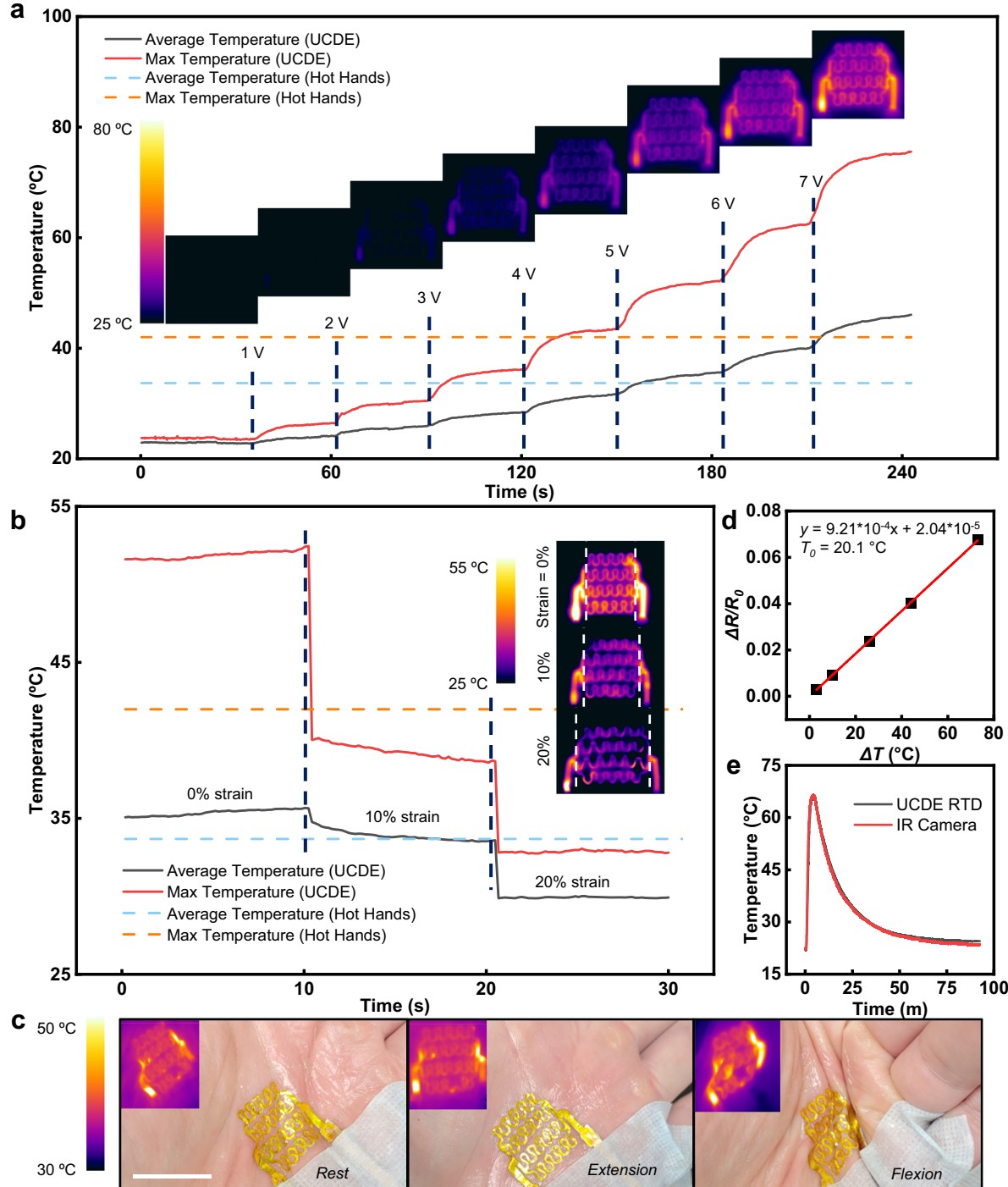

**Fig. 3 Stretchable heater and RTD sensor developed from UCDEs. a** The temperature evolution of the UCDE heater from 1 to 7 V DC bias voltage at 0% strain compared to commercially available hand warmers. Insets are the thermal profiles emitted from the UCDE heater, captured by an IR camera at corresponding voltages. **b** Characterization of the temperature output from the UCDE heater under deformation at 0, 10, and 20% strain. Insets are IR images of the UCDE heater under various strain deformations. **c** The stretchable UCDE heater is laminated on a palm and subjugated to commonplace hand deformations (rest, extension, and flexion). Insets are the thermal profiles emitted from the UCDE heater while laminated on the palm (scale bar, 2.5 cm). **d** The calibration curve of the UCDE RTD sensor, 4-probe resistance vs. temperature response of a thermocouple. **e** The temperature response of the UCDE RTD sensor compared with an IR camera.

electron transfer and interfacial properties of the UCDE electrodes were evaluated by cyclic voltammetry (CV) and electrochemical impedance spectroscopy (EIS), tested in phosphate-buffered saline (PBS) (pH 7.4) with 5 mM $K_3Fe(CN)_6$ (Fig. 4a–c). As shown in Fig. 4a, b, once the UCDE electrodes were electrochemically

cleaned in 0.1 M $H_2SO_4$, the redox reaction of the electroactive molecules was superior after electrochemical cleaning and performed similarly to a bare gold electrode (Supplementary Fig. 16). The UCDE's electrode performance before and after electrochemical cleaning was analyzed by EIS. Figure 4c presents the EIS

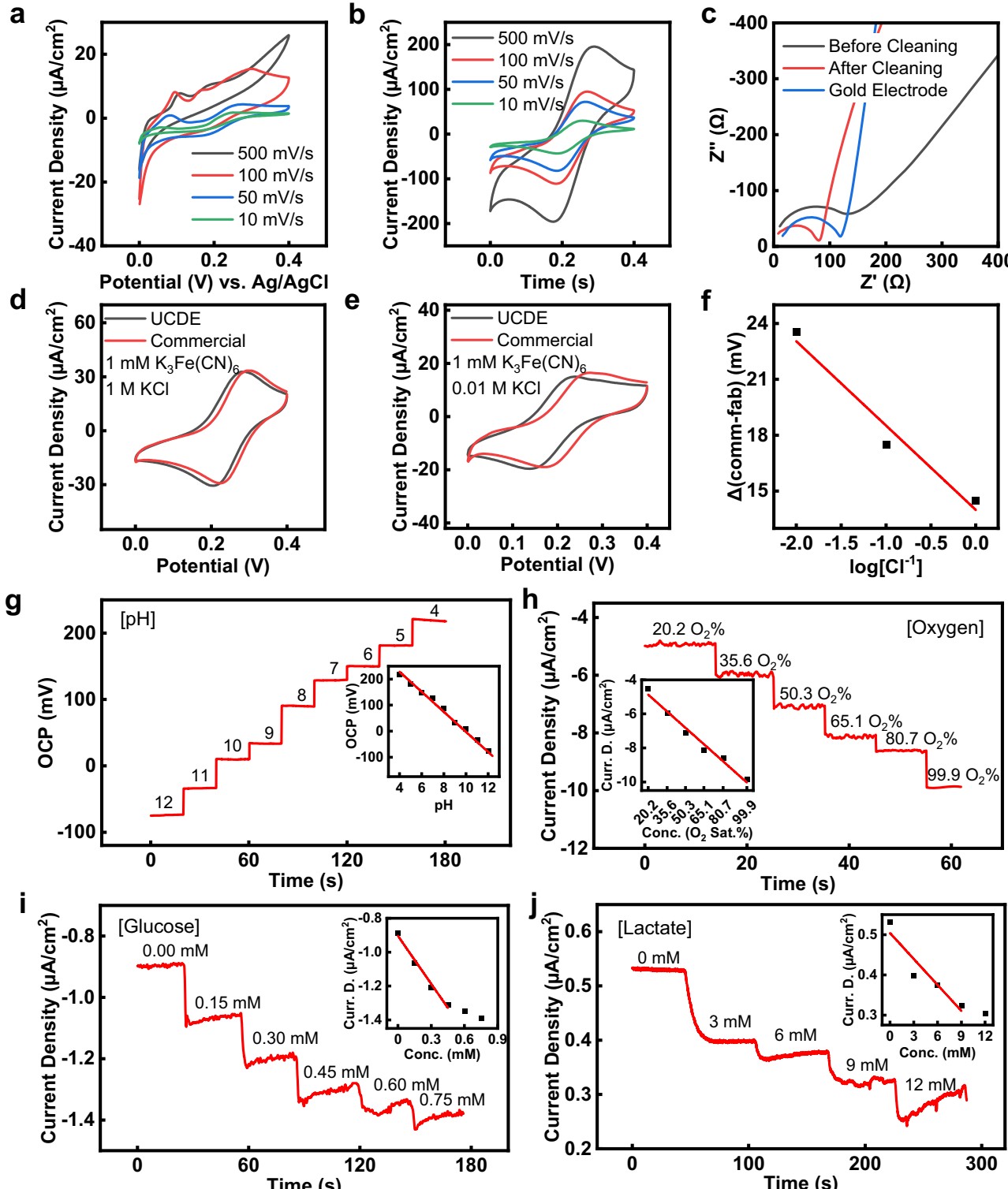

**Fig. 4 Characteristics of the stretchable, electrochemical UCDEs.** Cyclic voltammetry of UCDEs vs. Ag/AgCl (1 M KCl) in PBS (pH 7.4) with 5 mM $K_3Fe(CN)_6$ **a** before and **b** after electrochemical cleaning in $H_2SO_4$. **c** EIS performance in PBS (pH 7.4) with 5 mM $K_3Fe(CN)_6$. Cyclic voltammetry performance of UCDEs as a Ag/AgCl reference electrode vs commercial Ag/AgCl (1 M KCl) electrode with **d** 1 M and **e** 0.01 M Cl⁻. **f** Calibration curve of the UCDE reference electrode compared to a commercial Ag/AgCl (1 M KCl) electrode. **g** Potentiometric performance of the UCDE as a pH sensor (vs. fabricated Ag/AgCl UCDEs reference electrode). Inset is the associated calibration curve of the pH sensor. Amperometric performance of the UCDEs as a **h** oxygen, **i** glucose, and **j** lactate sensor (vs. fabricated Ag/AgCl UCDEs reference electrode). Insets are the corresponding calibration curves.

results of the UCDE and bare gold electrodes. The cleaned UCDEs and bare gold electrodes had a lower internal resistance and reactance, indicating the electrochemical redox-active site on the UCDEs become superior after acid cleaning.

Given that the UCDEs are composed of Ag and Au, these electrodes can be translated into highly functioning Ag/AgCl reference electrodes through a simple electrochemical process. Without electrochemical cleaning, linear sweep voltammetry (LSV) followed by CV in 0.1 M KCl and 0.01 M HCl produced nucleation of the AgCl, converting it into a conformal film atop the UCDEs electrodes[57]. The performance of the UCDE Ag/AgCl reference electrode compared to a commercial Ag/AgCl (1 M KCl) reference electrode is shown in Fig. 4d–f and Supplementary Table 2. As expected, with a decrease in Cl$^-$ concentration, the fabricated reference electrode presented a slight negative potential drift compared with the commercial Ag/AgCl (1 M KCl) reference electrode (Fig. 4f). Thus, this fabricated UCDE reference electrode could be used in place of a commercial Ag/AgCl electrode with a negligible change in performance.

Potentiometric, amperometric, and enzymatic UCDE sensor performance was monitored separately in different analyte solutions. The measurement of H$^+$ levels is needed to develop a pH sensor. We coupled the Ag/AgCl reference electrode with an H$^+$-selective ionophore embedded in a polyvinylchloride (PVC) coated membrane. Figure 4g shows the representative open circuit potential (OCP) response of the pH sensor, measured potentiometrically in solutions of 4–12 pH. The ISE showed a near-Nernstian cationic slope (Nernstian equation, theoretical sensitivity of ISE-based sensors is 59 mV/decade) with a sensitivity of −36.5 mV/decade ($R^2 = 0.99$) of concentration for H$^+$ ions was observed. Healthy pH values on the skin range from 4–7 pH, whereas a more basic pH on a wound can indicate a diseased state[14]. A Clark-type oxygen sensor was based on the interaction of Nafion and a diluted PDMS layer (oxygen selective membrane) coating the UCDE's electrode following electrochemical cleaning. The CV response showed a redox potential for oxygen at −0.4 V vs. UCDEs Ag/AgCl (Supplementary Fig. 17a). Figure 4h illustrates the chronoamperometric response of the oxygen sensor, capable of detecting dissolved oxygen concentrations between 20.2–100% O$_2$ saturation, well within physiological concentrations of blood 1.60 to 4.16 mg/L (10.5–27.7 O$_2$%)[58,59]. The UCDE oxygen sensor exhibited a sensitivity of −65 nA/(cm$^2$O$_2$%) ($R^2 = 0.98$) and 42 s response time ($t_{90\%}$) (Fig. 4h and Supplementary Fig. 17b). The sensing of glucose and lactate is based on glucose and lactate oxidase enzymes that are immobilized by a single-walled carbon nanotube (SWCNT)-chitosan solution on a Prussian Blue mediator layer[58,60]. After electrochemical cleaning, the Prussian Blue mediator layer was electrochemically deposited by CV. A five cycles CV deposition of Prussian Blue yielded an H$_2$O$_2$ response, shown in Supplementary Fig. 18a, b, presenting a dynamic range of 5 to 30 mM with a sensitivity of −1.85 µA/cm$^2$mM ($R^2 = 0.99$), which can be modified with a tradeoff of increased sensitivity (fewer CV cycles of Prussian Blue) or increased dynamic range (more CV cycles of Prussian Blue). Figure 4i, j shows the chronoamperometry response of the UCDE enzyme-based glucose and lactate sensors. The cyclic voltammetry response of the amperometric glucose and lactate sensors with the Prussian Blue mediator layer is presented in Supplementary Fig. 18c. The UCDE's glucose sensor produced a linear range between 0.15 to 0.75 mM at a sensitivity of −0.94 µA/cm$^2$mM ($R^2 = 0.98$), with physiologically relevant concentrations for sweat glucose levels, 0.2 to 0.6 mM[61]. The UCDE's lactate sensor demonstrated a linear range from 3 to 9 mM with a sensitivity of −21.5 nA/cm$^2$ mM ($R^2 = 0.98$), falling within healthy physiological concentrations between 1 to 3 mM

and >7 mM indicating lactic acidosis at a wound[14]. UCDE electrodes can be simply functionalized into fully developed potentiometric, amperometric, and enzymatic-based sensing systems, an inexpensive and rapid alternative to microfabrication, screen printing, and inkjet technologies.

**Upcycled CD electronics as a biodegradable resistor.** In addition to the development of physical sensors with the UCDEs, this upcycling process can be modified to produce biodegradable electronics, which have numerous clinical applications[62–64]. The CD composition presents an ultrathin layer of Au-Ag that can be easily exploited into biodegradable electronics. The UCDEs can be translated into biodegradable devices, by slightly changing the fabrication process (Supplementary Figs. 19, 20) and soaking them in nitric acid instead of acetone to fully remove the protective PMMA layer. The device consists of a passive biodegradable membrane (~50 µm thick) of polyvinyl alcohol (PVA) or polycaprolactone (PCL) with the active gold transferred from the CD (18.96 ± 5.28 nm thick) (Fig. 5a). The transient mechanism of PVA relies on the simple dissolution of the polymer substrate, whereas PCL can be degraded via hydrolytic degradation from PCL to 6-hydroxycaproic acid through hydrolysis (Fig. 5b)[65]. Biodegradable resistive-based sensors were fabricated with PVA and PCL substrates. The evaluations of the PVA and PCL substrates established quantitative metrics for the development and translation of these inexpensive, resorbable devices for use in clinical care and dissolve to yield completely biocompatible products. Because of the fast kinetics of the PVA device, it can be used as a rapid measurement sensor where removal is unnecessary (e.g., quick wound assessments) and the longer dissolution kinetics of the PCL-based device for implantable sensors. Additionally, the PVA configuration produced high transmittance levels (Supplementary Fig. 21). The electrical performance of the PVA-based device was terminated within less than a second in water but sustained in organic solvents (Fig. 5c). The PCL-based device presented an antithetical electrical response as the electrical performance was unperturbed in water but disturbed in organic solvents (Fig. 5d). In addition, the PCL device exhibited stable performance within various pH solutions (Fig. 5e). The various stages of dissolution are illustrated in Fig. 5f, g for the PVA and PCL resistor within biological conditions (PBS, 7.4 pH at 37 °C). Figure 5h, i and Supplementary Fig. 22 present the nanoscale dissolution of the PCL resistor observed by SEM, illustrating the PCL-metal interface. PCL has been demonstrated to degrade slowly in aqueous solutions and can take months to fully degrade (Supplementary Fig. 22)[65]. The PCL dissolved uniformly without fractures and the metal layer developed microcracks over time. Nonetheless, this device configuration can remain functional for months. In our study, we examined the resistance changes for 7 days with alternating temperature, which presented an increase in base resistance from 36 to 426 Ω (Supplementary Fig. 23).

**Biocompatibility of the upcycled CD electronics.** In vitro biocompatibility of skin keratinocyte cells (HaCaT) was evaluated on the UCDEs produced by various preparation procedures that involved soaking the CD within acetone, hydrochloric acid, or nitric acid. The five sample groups ($n = 3$) evaluated were control, acetone soak (Ac), hydrochloric acid soak (HCl), nitric acid soak (NA), and the gold flakes produced through transience as the electronics disassemble. After 7 days in culture on the experimental substrates, cell viability was assessed utilizing a live/dead assay (Fig. 6). HaCaT surrounding all substrates showed a confluent monolayer with high viability, as confirmed using fluorescent microscopy and fluorescent intensity analysis (Fig. 6a). The cells in the control group exceeded the measurable

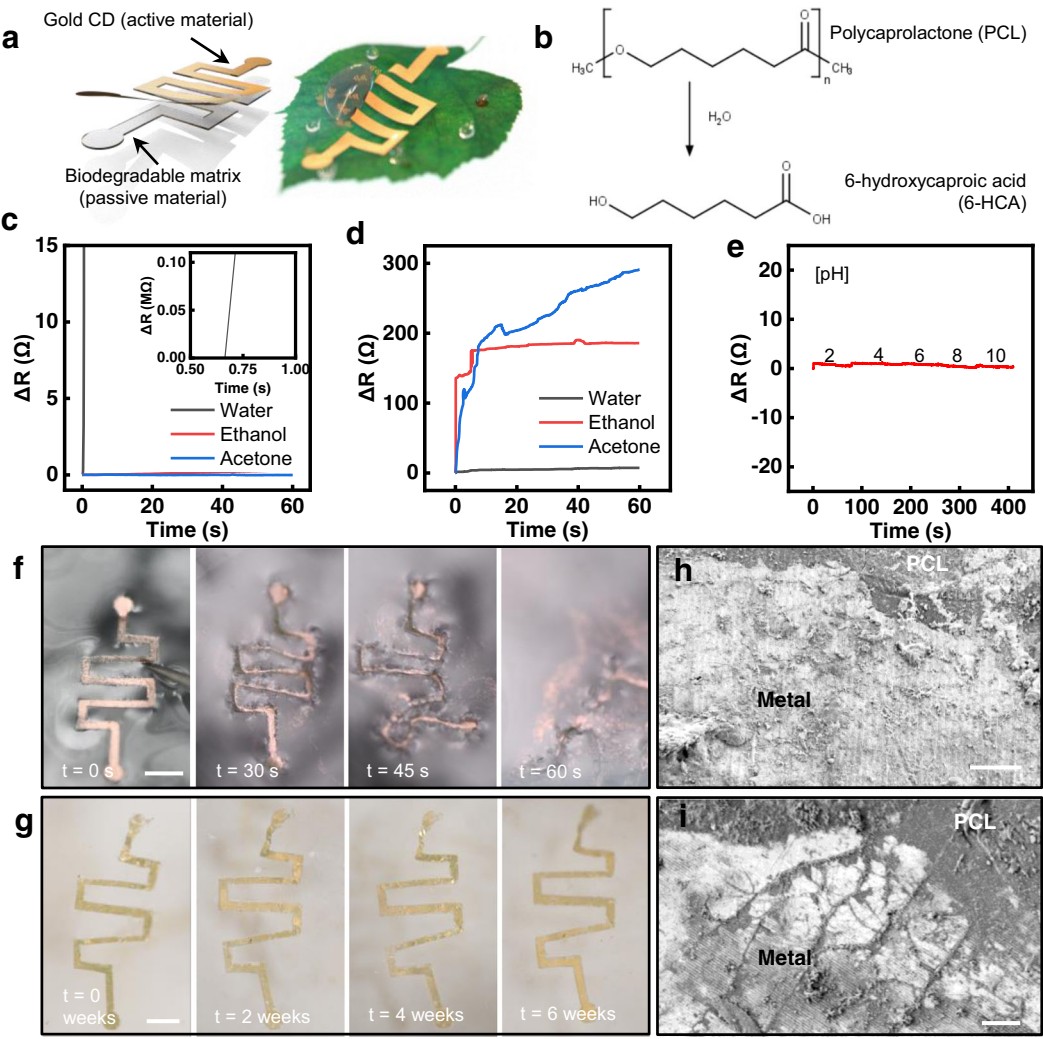

**Fig. 5 Moisture-triggered performance of the UCDEs as a biodegradable resistor. a** Schematic design, enabling biodegradable electronics for fully recyclable devices. **b** Chemical reaction responsible for triggering transience. **c** PVA-based, electrical degradation performance in various solvents. **d** PCL-based, electrical degradation in various solvents and **e** stability in different pH solutions. Degradation vs. time images in PBS (pH 7.4) of the **f** PVA and (scale bar, 3 mm) **g** PCL-based resistor (scale bar, 3 mm). SEM of metal-PCL interface: **h** before degradation and **i** after 6 weeks soaked in PBS (7.4 pH) at 37 °C (scale bar 10 μm).

fluorescent intensity for viable cells, where all three samples were fully confluent (Supplementary Fig. 24). Cells remained viable in groups Ac (~96.7%), HCl (~94.7%), and NA (~93.0%) after 7 days, while cells exposed to the gold flakes demonstrated statistically less viability (~77.8%) in vitro (*$p < 0.05$) (Fig. 6b and Supplementary Fig. 24b). We suspect the large size of the flakes (SA = ~110 μm) disrupted the natural motility and environment of the cells, preventing them from fully attaching and proliferating within these conditions. We hypothesize that in vivo, multinucleated macrophages, multinucleated giant cells, or foreign body giant cells would be able to clear out these flakes through phagocytosis at the expense of an elevated inflammatory response[66,67].

## Discussion

To date, researchers have presented methods to recycle CD waste into electrochemical sensors for scalable and cheap protocols. However, thus far, they have failed to demonstrate mechanically durable biosensor platforms for practical wearable applications. Our study addresses this limitation. These CDs can be transformed into soft bioelectronics for noninvasive monitoring, while fully

integrating with human skin. A mechanical machine cutter carefully defined the UCDEs for affordable micropatterning of fully stretchable and flexible electronics. We introduce a new upcycling approach and applications for fully scalable biopotential (EMG and ECG) sensing, heat emittance, temperature detection, electrochemical monitoring (pH, oxygen, lactate, and glucose), and moisture-triggered transient sensing. Biopotential sensors developed showed similar performance to commercially available gel electrodes. The heater demonstrated produced an average heat output of 35.6 °C at 5 V and the RTD sensor exhibited analogous sensitivity to a laboratory-based infrared camera. Potentiometric pH sensing illustrated a dynamic range from 4–12 pH and sensitivity of −36.5 mV/decade. The amperometric sensor performance for the oxygen sensor was 20.2–100 O$_2$% (sensitivity = −65 nA/(cm$^2$O$_2$%), glucose sensor was 0.15–0.75 mM (sensitivity = 0.94 μA/cm$^2$mM), and lactate sensor was 3–9 mM (sensitivity = −21.5 nA/cm$^2$ mM). Lastly, transient performance was demonstrated for fully recyclable electronics. This translational development was fully optimized, to produce biologically relevant results in stretchability and flexibility as well as sensing performance while remaining fully biocompatible. Overall, this study provides a useful

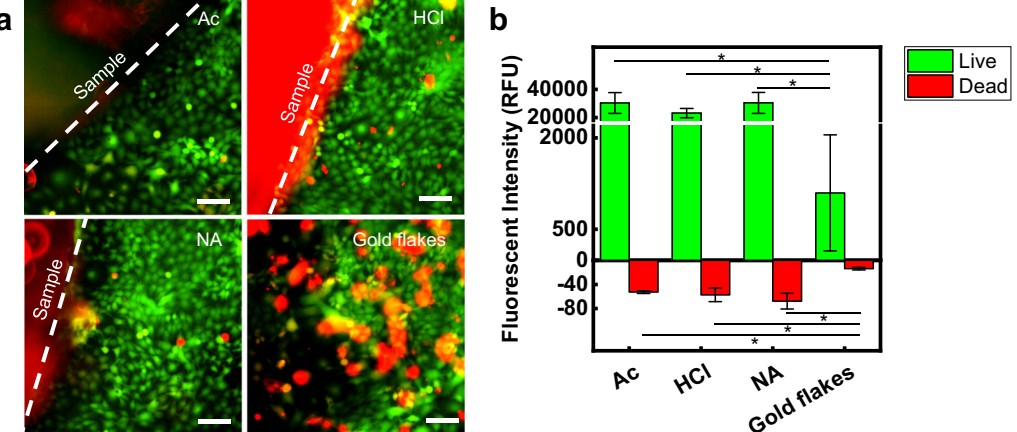

**Fig. 6 Biocompatibility of UCDEs. a** Confocal imaging of live/dead stained HaCaT cell cultured for 7 days (scale bar, 100 μm). The sample group of a soaking method for UCDEs: Acetone (Ac), Hydrochloric acid (HCl), Nitric acid (NA), and gold flakes. **b** Relative fluorescence intensity of cells cultured for 7 days is presented as average and standard error of means (*$p < 0.05$ TTEST).

alternative for e-waste management, one-time use electronics, rapid prototyping, and inexpensive approaches for bioelectronic fabrication methods.

The proposed upcycling process reported here enables sustainable solutions for CDs and other e-waste recycling which can be demonstrated in the future. We foresee additional work towards (1) evaluating the long-term performance of the electrochemical sensors, (2) fully integrated systems for wireless monitoring, and (3) additional studies to fully realize the potential of the transient devices in applications towards implantable bioelectronics. The upcycling method provided here will allow for bioelectronic fabrication without the need for intensive training and microfabrication techniques, which opens the door to a wider variety of disciplines adopting stretchable and flexible based devices for their studies.

Our paper illuminates the challenges that plague e-waste recycling and, subsequently, provides a remedy. Consumer confusion and a lack of infrastructure knowledge remain critical issues with regard to proper recycling. Emphasizing the development of novel upcycling approaches through scientific dissemination will increase awareness within this area. New programs incorporated into the Infrastructure Investment and Jobs Act, specifically the RECYCLE Act, aim to relieve the inundated recycling stream and provide new opportunities to support innovative recycling and upcycling ideas. Thus far, recycling and upcycling activities have accounted for 681,000 jobs, $37.8 billion in wages, and $5.5 billion in tax revenues in the United States[68]. Policy programs that fund new jobs and ideas will assist the United States to attain global sustainability objectives. Upcycling is a sustainable practice as it "meets the needs of the present without compromising the ability of future generations to meet their own needs"[69] through the transformation of waste into secondary products. The proposed upcycling approach remains sustainable if the cost of microfabrication remains exuberant, rapid prototyping persists as an essential business and institutional practice, and healthcare increasingly demands one-time use sensors. As a simple and cost-effective method, this technology can be adopted on both an academic research and commercial scale. Institutes and Universities may install CD collection boxes, while companies that provide CD collection methods, such as GreenDisk, may adopt or outsource the proposed fabrication techniques as an alternative to alleviate the accumulation of CDs in landfills. Any effort toward increasing both recycling and upcycling will advance the establishment of environmentally sustainable practices.

## Methods

All experiments with human subjects were performed in compliance with protocols that were approved by the Institutional Review Board at Binghamton University (IRB ID: STUDY00003602).

**Measurements and testing**. Verbatim archival gold CDs and the PI tape were purchased through Amazon for the UCDE fabrication illustrated in Fig. 1a and patterned with a Cricut Maker® fabric cutter. The contact pads of the UCDEs were bridged and connected with standard wires by a two-component electrically conductive silver epoxy. Parts A and B of the epoxy were mixed at equal ratios in weight and then placed onto the contact pad to electrically connect the lead wires. The silver epoxy was cured at 100 °C for 5 min. The UCDEs were connected to a digital multimeter (Keysight, 34460 A) for real-time measurements. Biopotential measurements were performed and processed with a PowerLab data acquisition unit and analyzed via LabChart software. Temperature images were captured in real-time by an infrared (IR) camera (ETS320). All electrochemical tests were performed with a potentiostat (CH Instruments, 660E).

**Mechanical testing**. All mechanical tests were performed with a group size of $n = 3$, and a Mark10 tensometer using a 25 N force gauge. Stress and strain testing were produced with a strain rate of 5.1 mm/min to failure. The strain rate for cyclic bending was 300 mm/min and held at a bending radius of 3.5 mm. All experiments were performed with a digital multimeter (Keysight, 34460 A) to record the real-time resistance.

**Microcontroller unit (MCU)**. The ECG MCU was designed with a uBIC-MZ24C20R (MEZOO, Inc, South Korea) chipset, which is a high-performance, low-powered one-chip 1 channel ECG (lead I) biometric sensor module with a 32-bit ARM Cortex-M0 processor. ECG data from two leads (RA and LA) were collected with 24-bit ADC resolution and 1 kHz sampling rate and then transmitted to a smartphone application in real-time via Bluetooth low-energy (BLE) communication.

**Electrochemical cleaning**. All electrodes (except the reference and pH electrode) were cleaned in 0.1 M $H_2SO_4$ from −0.4 to 1.4 V (vs. Ag/AgCl (1 M KCl)) at 25 mV/s for one cycle.

**Reference electrode**. The reference electrode was fabricated by utilizing the trace amount of silver within the active electrode material from the CD. The Ag was chlorinated in an aqueous solution of 0.1 M KCl and 0.01 M HCl with linear sweep voltammetry from open circuit potential (OCP) to 0.4 V (vs. Ag/AgCl (1 M KCl)) at 20 mV/s followed by cyclic voltammetry from 0.1 to 0.3 V (vs. Ag/AgCl (1 M KCl)) at 100 mV/s for ten cycles[57].

**pH electrode**. The fabricated reference electrode was used for the development of the pH sensor with a pH-sensitive membrane coating the Ag/AgCl electrode. The pH ISE solution was prepared with 1% (v/v) $H^+$ ionophore I, 0.1 wt% potassium tetrakis(4-chlorophenyl)borate, 10% (v/v) nitrophenyl octyl ether, and 5 wt% polyvinylchloride (PVC) in tetrahydrofuran. A 3 μL solution was drop cast on the Ag/AgCl electrode.

**Oxygen electrode**. The oxygen sensor was prepared by drop-casting three layers of 3 μL of Nafion onto the gold electrode and allowing each layer to dry for an hour. A selective diffusion membrane was drop cast at 3 μL which contained 30 wt % of PDMS in toluene. The drop cast mixture was then cured at 60 °C for 1 h.

**Lactate and glucose electrode**. The immobilization solution (chitosan/SWCNT) was prepared by mixing 2% acetic acid with 1% chitosan in deionized water and stirred for 2 h. Next, SWCNTs were added at a loading density of 2 mg/mL of solution and water bath sonicated for 30 min. The Prussian Blue mediator layer was electrochemically deposited in a fresh solution of 100 mM KCl, 2.5 mM $K_3Fe(CN)_6$, 2.5 mM $FeCl_3$, and 100 mM HCl. For the lactate sensor, the Prussian Blue mediator layer was electrochemically deposited through cyclic voltammetry from −0.5 to 0.6 V (vs. Ag/AgCl (1 M KCl)) at 50 mV/s for five cycles. After deposition, the electrodes were rinsed with DI water and 3 μL of the chitosan/SWCNT solution was drop cast onto the electrode and allowed to dry for 1 h. Lactate oxidase solution (40 mg mL$^{-1}$ in PBS (pH 7.4)) was drop cast at 2 μL and allowed to dry for an hour. Finally, another 3 μL of the chitosan/SWCNT solution was drop cast onto the electrode and allowed to dry for 1 h. The electrode was stored overnight in a refrigerator. For the glucose sensor, the Prussian Blue mediator layer was electrochemically deposited by cyclic voltammetry from 0 to 0.6 V (vs. Ag/AgCl (1 M KCl)) at 25 mV/s for one cycle. Glucose oxidase solution was prepared and mixed (10 mg mL$^{-1}$ in PBS (pH 7.4)) and added to the mixture of chitosan/SWCNT at a ratio of 1:2 (volume by volume). The glucose oxidase solution was drop cast at 3 μL onto the electrode and allowed to dry for 1 h. Then a 3 μL solution of chitosan/SWCNT was drop cast atop and allowed to dry for 1 h and then placed in the refrigerator overnight.

**Cell culture**. All samples were UV-sterilized for 30 min and attached to a tissue culture plate. HaCaTs, immortalized keratinocyte cells derived from human skin, were grown in Dulbecco's Modified Eagle media supplemented with 10% fetal bovine serum and 1% penicillin-streptomycin antibiotics. Passage 8 HaCaT were seeded at 60,000 cells/sample and media was replenished every 48 h, where gold flakes were also replenished in corresponding sample wells. Cells were cultured for 7 days on all substrates until a live/dead assay was performed using 3 μM Calcein AM and 3 μM Propidium Iodide. Imaging was conducted using a fluorescent microscope (Nikon) and fluorescent intensity was obtained using a plate reader (Tecan).

**Statistics and reproducibility**. No statistical method was used to predetermine sample size. Data in Figs. 1g–i, 6b and Supplementary Fig. 12b is presented as the average and standard error of means with a group size of $n = 3$.

**Reporting summary**. Further information on research design is available in the Nature Research Reporting Summary linked to this article.

## Data availability

All relevant data supporting the fabrication, testing, and functionalization of the sensors within the study are presented in the paper and Supplementary Information file. Additional information may be requested from the corresponding author upon reasonable request. Further details are described within the Supplementary Information file that includes an alternative fabrication method, temperature sensor calibration, electrochemical performance evaluation, the fabrication of the biodegradable UCDEs, and solutions to recycle acetone and polycarbonate.

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

## Acknowledgements

General: We thank the staff of the Nanofabrication Facilities (NLB) and Analytical and Diagnostics Laboratory (ADL) at Binghamton University for technical support. We appreciate the support from Chae Ho Cho and Ajan Prabakar in developing the smartphone application for wireless ECG recording. We also would like to thank Sean McGee from the Binghamton University Environmental Studies Program for supporting our research with regard to environmental policy and sustainability. This work was supported by the National Science Foundation (ECCS #2020486 and #1920979). We acknowledge the support of the Small-Scale Systems Integration and Packaging Center of Excellence (S3IP), BU-UHS Seed Grant Funding, and Start-up funds at SUNY Binghamton.

## Author contributions

M.S.B. and A.K. led the development idea and designed the experiments. M.S.B. performed the experiments and wrote the paper. L.S. conducted the mechanical testing experiments. M.M. conducted the biocompatibility studies and contributed to writing the corresponding section, while G.J.M. provided guidance. Y.N. developed the Bluetooth MCU and associated software. A.K. supervised this work, provided guidance, and assisted in drafting the manuscript as the corresponding author.

## Competing interests

The authors declare no competing interests.
