## [Peer Review File · Nature Communications]

Upcycling compact discs for flexible and stretchable bioelectronic applicationsReviewers' comments:

Reviewer #1 (Remarks to the Author):

The article is interesting. The results are very well presented and also well discussed. The text is also complete in terms of images and diagrams that make the reading fluent and effective. The sensor preparation method is certainly useful and interesting also from the point of view of the circular economy. The main problem with this article, however, is originality. In the literature there are numerous examples of the reuse of CDs/DVDs for the production of sensors with methods similar, if not in some cases the same as the one proposed here. In particular, the authors are advised to read the following articles, to include them in the introduction and to underline their novelty and the advantages / disadvantages with respect to their method.

10.1109/FLEPS51544.2021.9469858

10.1021/ac000437p

10.1016/j.aca.2021.339215

10.5923/s.aac.201307.05

10.1109/NEBC.2009.4967654

10.13140/RG.2.1.2477.2005

10.1016/j.trac.2017.04.013

10.1016/j.bios.2012.08.032

10.3390/mi9040187

10.1016/j.cirpj.2015.01.005

Reviewer #2 (Remarks to the Author):

The authors reported an upcycling process that enables sustainable solutions for CDs and other e-waste recycling. These CDs can be transformed into soft bioelectronics for noninvasive monitoring, while fully integrating with human skin and that can communicate with a smartphone via Bluetooth. This paper is a very interesting, well written, complete with all necessary information in supplementary section and suitable for publication in Nature Communications.

Reviewer #3 (Remarks to the Author):

My comments on:

Upcycling Compact Discs for Bioelectronic Applications

Matthew S. Brown , Louis Somma, Melissa Mendoza, Yeonsik Noh, Gretchen J. Mahler, and Ahyeon Koh^{1*}

General comments:

In this manuscript the authors describe a procedure to prepare wearable gold electrodes with the material from CD- DVD . As pointed by the authors, this is very important from the point of view of environmental issue and e-waste .

The aim of the work stated by the authors is "This sustainable approach for upcycling e-waste provides an advantageous research-based waste stream that does not require cutting-edge microfabrication facilities, expensive materials, and high-caliber engineering skills." The approach is absolutely correct, but I would like to have a comment by the authors on the feasibility and the real balance of the amount of e-waste consumed in the proposal procedure. Many brilliant proof-of-concept of sensor remain not applicable from technological and commercial point of view.

The authors state that the "Existing microfabrication techniques for fabricating stretchable, active components have primarily relied on costly and time-consuming printing or lithography-based technologies". It would be necessary a comment from the authors to compare technologically the real improvement from environmental and economical aspect s.

It is true that "One-time-use, disposable sensors are in growing demand for reliable, accessible, and fast measurements, and that can be used anywhere or any time without recalibration or the worry of contamination" , but the realization requires a huge amount of work. It is not clear how long is the time of preparation of these electrodes respect to the usual procedures.

Acetone is not a real green solvent and several acids to. How much will be consumed in the fabrication of these electrodes?

The work is well done and really sounds scientifically. The approach and strategy are correct and data are well analysed, apart some questions in the electrochemical part. However, the idea of using the gold "lines" present in CD and DVD to fabricate electrodes is not new. The first application was published 20 years ago by L. Angnes et al. Gold Electrodes from Recordable CDs, Anal. Chem. 2000, 72, 5503-5506. The electrodes produces were not wearable, but the precious material was recycled and found application is many ways, as indicated in a recent review by G.Moro et al. "Disposable electrodes from waste materials and renewable sources for (bio) electroanalytical applications" Biosensors and Bioelectronics 146 (2019) 111758, <https://doi.org/10.1016/j.bios.2019.111758> .

The present manuscript is a good work and idea, but it represents an improvement of old ideas, and I think that it could be mentioned..

The quality of the data is good.

The level of support for the conclusions for the electroanalytical part needs to be improved . Some analytical study should be complemented.

The use of e-waste, particularly from CD/DVD is not a novelty, but the technology proposed for the fabrication of wearable sensors is a good novelty.

Some particular comments:

Line 30:... "from the CD was $30.35 \pm 1.92 \mu\text{m}$, consisting of a protective, polymethylmethacrylate"... In the paper mentioned above Angnes indicated 50-100 nm for this thickness. How was it measured?

Lines 42-47: . "Energy dispersive X-ray spectroscopy (EDS) analysis of the metal layer after the solvent treatments are shown in Figure 1F and S9. After the soaking in acetone, Ag and Au could be seen within the spectrum at 70.95 and 29.05 wt.%, respectively (Supplementary Fig. 9A-B). Their presence confirmed the archival composition of the layer as predominantly Ag. Additional methods to treat the CD are discussed in the Supporting Information. The CD metal layer can be stripped down to nearly pure gold by soaking in a bath of nitric acid." However, on lines 296-297, the authors mention that for the construction of the reference electrode it is possible to use the TRACE amounts of Ag present within the electroactive material of the CD. There is some incoherence. Could the authors clarify this point?

Lines 69-73: "A fully fabricated UCDE device consisted of two biopotential electrodes, a heater or temperature sensor, a reference electrode, a counter electrode, a pH electrode, an oxygen electrode, a lactate electrode, and a glucose electrode (Fig. 1K). The full, end-to-end fabrication and manufacturing required resources that can be found easily at conventional craft stores, negating the need for high-end instrumentation. "

The authors say that the fully UCDE is composed by 2 biopotential electrodes , but immediately describe that there are electrode to measure: pH, oxygen, lactate and glucose (4). Please clarify this spoint.

Lines 178-180: "A Clark type oxygen sensor was based on the interaction of Nafion and a diluted PDMS layer (oxygen selective membrane) coating the UCDE's electrode following electrochemical cleaning". How is the oxygen sensor calibrated? Usually Clark electrode requires frequent calibration.

Lines 196-201: "The UCDE's glucose sensor produced a dynamic range between 0.15 mM to 0.75 mM at a sensitivity of $-0.94 \mu\text{A}/\text{cm}^2\text{mM}$ ($R^2 = 0.98$) and limit of detection, 0.75 mM, with physiologically relevant concentrations for sweat glucose levels, 0.2 to 0.6 mM . The UCDE's

lactate sensor demonstrated a dynamic range from 3 to 9 mM with a sensitivity of $-21.5 \text{ nA/cm}^2 \text{ mM}$ ($R^2 = 0.98$) and limit of detection, 12 mM ,..."

Here there is a misunderstanding on the concept of "limit of detection". According to IUPAC: The limit of detection, expressed as the concentration, c_L , or the quantity, q_L , is derived from the smallest measure, x_L , that can be detected with reasonable certainty for a given analytical procedure. The value of x_L is given by the equation $x_L = \bar{x}_b + k s_b$ where \bar{x}_b is the mean of the blank measures, s_b is the standard deviation of the blank measures, and k is a numerical factor chosen according to the confidence level desired. Probably the authors interpreted this values as the linear range or the dynamic range .

(IUPAC. Compendium of Chemical Terminology, 2nd ed. (the "Gold Book"). Compiled by A. D. McNaught and A. Wilkinson. Blackwell Scientific Publications, Oxford (1997). Online version (2019-) created by S. J. Chalk. ISBN 0-9678550-9-8. <https://doi.org/10.1351/goldbook>.)

Another aspect not clear is how do the authors intend to do the quantification of the analyte, since it is not possible to calculate the concentration from a calibration plot obtained from standard additions of the analytes, as presented in the work. This is an tricky point common to almost all electrochemical sensors. The authors did not gave information about the reproducibility of the fabrication and well as of the measurements. It is necessary to have a statistical study on the performance of the sensors.

Lines 253-255: "We hypothesize that in vivo, multinucleated macrophages, multinucleated giant cells, or foreign body giant cells would be able to clear out these flakes through phagocytosis at the expense of an elevated inflammatory response ". This hypothesis should be confirmed .

Lines 294-295: "Electrochemical Cleaning: All electrodes (except the reference and pH electrode) were cleaned in $0.1 \text{ M H}_2\text{SO}_4$ from -0.4 V to 1.4 V (vs. Ag/AgCl (1M KCl)) at 25 mV/s for 1 cycle." The cleaning of gold electrode surface is very tricky and important. Usually just one CV cycle is not enough to clean a gold electrode. See: Campos et al, <https://doi.org/10.1016/j.electacta.2013.07.083> for cleaning a gold electrode.

Lines 296-297: "Reference Electrode: The reference electrode was fabricated by utilizing the trace amount of silver within the active electrode material from the CD."

Here the authors indicate that the material of the CD useful as active electrode contains trace amount of silver. See a previous comment.

Probably many American researchers use hr and hrs to represent hour and hours. However the international abbreviation is h. I consider that it is important to respect international rules.

Lines 313-315: "The solution of Prussian Blue consisted of 100 mM KCl , $2.5 \text{ mM K}_3\text{Fe}(\text{CN})_6$, 2.5 mM FeCl_3 , and 100 mM HCl . For the lactate sensor, the Prussian Blue mediator layer was electrochemically deposited through cyclic voltammetry from -0.5 V to 0.6 V (vs. Ag/AgCl)."

It is well known that Prussian Blue is not soluble in water, so you cannot prepare a "solution of Prussian Blue". What is indicated in this text is the solution present in the cell for the electrochemical deposition of PB. Please correct.

Reviewer #4 (Remarks to the Author):

The title of your manuscript is interesting given the importance of upcycling the waste resources within our technosphere. However, I find some bigger challenges not addressed by this manuscript.

First of all, as you have mentioned yourself in line 36 that the compact discs are dated technologies, then why spend precious resources on figuring out a waste treatment option for them. An estimated 5.5. million CDs are discarded currently, but what about coming years when they wouldn't appear anymore in our waste flows. Already, we don't see them in noticeable number/amount in e-waste pre-processing facilities in many countries. In such a case, are you looking to dig up the urban mines (landfills) to get the CDs for your upcycling process? and how environment friendly and sustainable that's going to be?

Secondly, you mentioned in line 33 that 15-20% of e-waste gets recycled despite its valuable materials. Let me clarify for you that valuable materials are the first resources recovered from e-waste irrespective of developed or developing nations. So, this 15-20% could be a representative of valuable materials found in e-waste, where the recycling companies are simply not interested in recovering the remaining bulk for economic reasons. So, rather than focusing on obsolete technologies, it would be good to devote all our energies to figure out ways for the remaining bulk materials, e.g., plastics and glass.

It would be good to add a table of input and outputs for the upcycling process to visualize how many resources are used to little benefit. Also, what about a constructive discussion on demand for bioelectronic applications?

Reviewer #1

Summary Recommendation

The article is interesting. The results are very well presented and also well discussed. The text is also complete in terms of images and diagrams that make the reading fluent and effective. The sensor preparation method is certainly useful and interesting also from the point of view of the circular economy. The main problem with this article, however, is originality. In the literature there are numerous examples of the reuse of CDs/DVDs for the production of sensors with methods similar, if not in some cases the same as the one proposed here. In particular, the authors are advised to read the following articles, to include them in the introduction and to underline their novelty and the advantages / disadvantages with respect to their method.

10.1109/FLEPS51544.2021.9469858

10.1021/ac000437p

10.1016/j.aca.2021.339215

10.5923/s.aac.201307.05

10.1109/NEBC.2009.4967654

10.13140/RG.2.1.2477.2005

10.1016/j.trac.2017.04.013

10.1016/j.bios.2012.08.032

10.3390/mi9040187

10.1016/j.cirpj.2015.01.005

Our response: We appreciate the time and effort taken to review our publication. We hope that given the additional changes we made to our manuscript, demonstrate our originality and significance, to reconsider your concerns.

We agree that there are previous examples of reusing CDs to produce electrochemical sensors. However, our paper is significant because it demonstrates transfer printing and harvesting active components from CDs, enabling electrochemical sensors by introducing enzymatic, Clark-type, and pH (potentiometric) sensing for biointegrated biomedical applications, which has yet to be demonstrated in literature. The previously reported research is unable to be use for on-skin biosensing applications and is limited to lab-based analysis due to rigid substrates. We highlight the development of a full 3-electrode system on ultrathin flexible and stretchable electronics. Our work exists on a stretchable and flexible platform, upcycled from CDs, and describes a completely integrated full 3-electrode system.

These articles only reference electrochemical sensors which, as described in the manuscript, accounts for only 20% of our study. We have yet to see published work that demonstrates our other sensing mechanisms (e.g., biopotential, resistive, and biodegradable sensors) on CDs.

We incorporated the previous studies that you recommended with a few additional studies.

The last paper 10.1016/j.cirpj.2015.01.005 used low powered laser cleaning to recover the polycarbonate, which we do not explore within our study because of the costly equipment required. Additionally, these researchers didn't upcycle any components following the cleaning process; therefore, we feel that this paper would be an irrelevant source to cite considering the method remains unproven for applicational uses.

Modifications to the manuscript: (Introduction, line 83–88) “To date, researchers have explored alternative uses for CDs to develop gold and silver electrodes^{33, 34, 35, 36}, detect metal ions (e.g., Pb, Hg, Cu, etc.)^{37, 38, 39}, screen organic compounds (e.g., DNA, cysteine, dopamine, etc.)^{40, 41, 42} and quantify oxidizing agents (e.g., hydrogen peroxide, Cl, iodine, etc.)^{43, 44, 45, 46, 47}. However, the techniques reported thus far fail to demonstrate an application pathway for biosensor platforms and lack the mechanical durability to be practical for wearable applications.”

(Reference updates):

33. Honeychurch, K. C. Cheap and disposable gold and silver electrodes: Trends in the application of compact discs and digital versatile discs for electroanalytical chemistry. *Trends Analyt. Chem.* **93**, 51-66 (2017).
34. Angnes, L., et al. Gold Electrodes from Recordable CDs. *Anal. Chem.* **72**, 5503-5506 (2000).
35. Cruz-Ramírez, A., et al. Progress on the Use of Commercial Digital Optical Disc Units for Low-Power Laser Micromachining in Biomedical Applications. *Micromachines* **9**, (2018).
36. Moro, G., et al. Disposable electrodes from waste materials and renewable sources for (bio)electroanalytical applications. *Biosens. Bioelectron.* **146**, 111758 (2019).
37. Honeychurch, K. Underpotential Deposition of Lead at Silver Electrodes Manufactured from Compact Discs and its Determination in Environmental Water Samples. *Adv. Anal. Chem.* **3**, 28-33 (2013).
38. Radulescu, M.-C. & Danet, A. F. Mercury Determination in Fish Samples by Chronopotentiometric Stripping Analysis Using Gold Electrodes Prepared from Recordable CDs. *Sensors* **8**, 7157-7171 (2008).
39. Richter, E. M., et al. Compact Disks, a New Source for Gold Electrodes. Application to the Quantification of Copper by PSA. *Electroanalysis* **13**, 760-764 (2001).
40. Cheng, H., et al. Disposable CD electrodes for DNA-based detection of *Ralstonia solanacearum*. In: 2009 IEEE 35th Annual Northeast Bioengineering Conference (2009).
41. Lowinsohn, D., et al. Disposable Gold Electrodes with Reproducible Area Using Recordable CDs and Toner Masks. *Electroanalysis* **18**, 89-94 (2006).
42. de Santana, P. P., de Oliveira, I. M. F. & Piccin, E. Evaluation of using xurography as a new technique for the fabrication of disposable gold electrodes with highly reproducible areas. *Electrochem. Commun.* **16**, 96-99 (2012).

43. Honeychurch, K. & Maynard, C. Amperometric determination of Hydrogen Peroxide at a Silver Electrode Fabricated from a Recycled Compact disc. *Adv. Anal. Chem.* **5**, 25-30 (2015).
44. Wen, Y., et al. From DVD to dendritic nanostructure silver electrode for hydrogen peroxide detection. *Biosens. Bioelectron.* **41**, 857-861 (2013).
45. Patella, B., et al. Electrochemical detection of chloride ions using Ag-based electrodes obtained from compact disc. *Anal. Chim. Acta* **1190**, 339215 (2022).
46. Patella, B., et al. Silver based sensors from CD for chloride ions detection. In: 2021 IEEE International Conference on Flexible and Printable Sensors and Systems (2021).
47. Cho, H., Parameswaran, M. & Yu, H.-Z. Fabrication of microsensors using unmodified office inkjet printers. *Sens. Actuators B Chem.* **123**, 749-756 (2007).

Reviewer #2

Summary Recommendation

The authors reported an upcycling process that enables sustainable solutions for CDs and other e-waste recycling. These CDs can be transformed into soft bioelectronics for noninvasive monitoring, while fully integrating with human skin and that can communicate with a smartphone via Bluetooth. This paper is a very interesting, well written, complete with all necessary information in supplementary section and suitable for publication in Nature Communications.

Our response: We thank the reviewer for the time they took to assess our manuscript and we value their response.

Summary Recommendation

In this manuscript the authors describe a procedure to prepare wearable gold electrodes with the material from CD- DVD . As pointed by the authors, this is very important from the point of view of environmental issue and e-waste .

Our response: We appreciate the reviewer's detailed comments to improve our manuscript. We hope that the changes we implemented in our revision are deemed acceptable.

- **Comment (1)** The aim of the work stated by the authors is “This sustainable approach for upcycling e-waste provides an advantageous research-based waste stream that does not require cutting-edge microfabrication facilities, expensive materials, and high-caliber engineering skills.” The approach is absolutely correct, but I would like to have a comment by the authors on the feasibility and the real balance of the amount of e-waste consumed in the proposal procedure. Many brilliant proof-of-concept of sensor remain not applicable from technological and commercial point of view.

The authors state that the “Existing microfabrication techniques for fabricating stretchable, active components have primarily relied on costly and time-consuming printing or lithography-based technologies”. It would be necessary a comment from the authors to compare technologically the real improvement from environmental and economical aspects.

It is true that “One-time-use, disposable sensors are in growing demand for reliable, accessible, and fast measurements, and that can be used anywhere or any time without recalibration or the worry of contamination”, but the realization requires a huge amount of work. It is not clear how long is the time of preparation of these electrodes respect to the usual procedures.

Our response: We agree that not many proof-of-concepts remain applicable from a commercial point of view, however, we believe that our study demonstrates an impactful approach at a first step towards upcycling e-waste. From a feasibility approach, two full devices can be created from one CD (illustrated in Figure 1J) on orders of magnitude cheaper and quicker than microfabrication techniques, costing around \$1.50 and taking between 20 to 30 mins per device. The procedure and feasibility are demonstrated in paragraph 1 of the main text.

We agree that the adoption of these processes would require a large scale of work and require more development to be fully implemented. We are presenting a first of its kind, case study. We will continue modifying and improving our upcycled devices to move towards the implementation of these types of sensors. But even so, stretchable bioelectronics are not currently available for real-world uses thus far, let alone one-time-use sensors. As a field, a lot of advancements are required to translate these developments to commercial markets.

In paragraph 2 of the Introduction, we added associated processing cost and lead time with microfabrication. We describe the time it takes to make these devices, around 20 to 30 minutes. In comparison to microfabrication, lithography-based techniques, ignoring the requirement of all the expensive equipment required, similar device fabrication can range from a few hours to days. Additionally, the increased cost associated with microfabrication wouldn't make sense for development of one-time-use sensor, since the production costs are significant.

With the advantages of our study, we believe the novel approach reported here can be readily used in small-scale applications, biomedical and other disciplines (e.g., anthropology, nursing, and psychology) for evaluating physiological information with on-body systems.

Modifications to the manuscript: (Introduction, line 69–73) “Processing cost significantly vary by facility, costing between \$2,702–\$7,298 per use and \$59,016–\$139,542 annually²⁶. The lead time can range from a few hours to days depending on the complexity of the device. Moreover, these processes require an abundance of volatile compounds (e.g., chemical etchant, photoresist, developer, etc.) that present environmental hazards²⁷”

(Reference updates):

26. Mahmood, A. & Reger, R. Microfabrication Process Cost Calculator. In: 2010 18th Biennial University/Government/Industry Micro/Nano Symposium (2010).

27. Chein, H. & Chen, T. M. Emission characteristics of volatile organic compounds from semiconductor manufacturing. *J. Air Waste Manag. Assoc.* **53**, 1029-1036 (2003).

- *Comment (2) Acetone is not a real green solvent and several acids to. How much will be consumed in the fabrication of these electrodes?*

Our response: Yes, we agree. Though acetone isn't necessarily a green solvent, it is one of the least hazardous industrial solvents when compared to photoresist, developer, and chemical etchants commonly used through lithography patterning. Additionally, multiple samples can be treated with an acetone bath which significantly reduces the required solvent volume per device.

To develop the main, upcycled electronics, acetone is the only solvent required. Therefore, our significance here is such that, for fabricating these devices we only require the use of acetone which is inexpensive and less toxic than photoresist, developer, and chemical etchants. For the transient device, diluted nitric acid is required. This part of the manuscript is purely a proof-of-concept, and we don't regard it as the main significance of our proposed study.

We added to the manuscript the volume of acetone utilized for soaking the CD.

Modifications to the manuscript: (Main Text, line 10–12) “The CD was soaked in 100 mL of acetone for 1.5 mins, releasing the metal layer by breaking down the polycarbonate substrate and dissolving the BPA (Fig. 1A1; Supplementary Fig. 4 and 5).”

- **Comment (3)** The work is well done and really sounds scientifically. The approach and strategy are correct and data are well analysed, apart some questions in the electrochemical part. However, the idea of using the gold “lines” present in CD and DVD to fabricate electrodes is not new. The first application was published 20 years ago by L. Angnes et al. Gold Electrodes from Recordable CDs, Anal. Chem. 2000, 72, 5503-5506. The electrodes produces were not wearable, but the precious material was recycled and found application is many ways, as indicated in a recent review by G.Moro et al. “Disposable electrodes from waste materials and renewable sources for (bio) electroanalytical applications” Biosensors and Bioelectronics 146 (2019) 111758, <https://doi.org/10.1016/j.bios.2019.111758>.

The present manuscript is a good work and idea, but it represents an improvement of old ideas, and I think that it could be mentioned.

The quality of the data is good.

The level of support for the conclusions for the electroanalytical part needs to be improved. Some analytical study should be complemented.

The use of e-waste, particularly from CD/DVD is not a novelty, but the technology proposed for the fabrication of wearable sensors is a good novelty.

Our response: We agree that there are previous examples of reusing CDs. We addressed this per Reviewer 1’s recommendations and made the appropriate changes to our manuscript by adding in additional references. Please look at our response to Reviewer 1. We have developed enzymatic, Clark-type, and potentiometric sensors, and a full 3-electrode system on a flexible and stretchable soft bioelectronics platform which have not been demonstrated with a CD thus far.

Our work isn’t solely describing electrochemical sensors. We have yet to see published work that demonstrates our other sensing mechanisms (e.g., biopotential, resistive, and biodegradable sensors) on CDs.

Our supporting electroanalytical studies are presented within the supplementary text of the manuscript.

- **Comment (4)** Line 30:... “from the CD was $30.35 \pm 1.92 \mu\text{m}$, consisting of a protective, polymethylmethacrylate” ... In the paper mentioned above Angnes indicated 50-100 nm for this thickness. How was it measured?

Our response: This was measured by SEM and presented in Supplementary Figure S7. As mentioned in the text, this thickness measurement includes the protective and metal layer from the CD. When the protective layer is removed, the thickness is decreased to ~70 nm which would match to what Angnes et al. has described.

- **Comment (5)** Lines 42-47: . “Energy dispersive X-ray spectroscopy (EDS) analysis of the metal layer after the solvent treatments are shown in Figure 1F and S9. After the soaking in acetone, Ag and Au could be seen within the spectrum at 70.95 and 29.05 wt.%, respectively (Supplementary Fig. 9A-B). Their presence confirmed the archival composition of the layer as predominantly Ag. Additional methods to treat the CD are discussed in the Supporting Information. The CD metal layer can be stripped down to nearly pure gold by soaking in a bath of nitric acid.”

However, on lines 296-297, the authors mention that for the construction of the reference electrode it is possible to use the TRACE amounts of Ag present within the electroactive material of the CD. There is some incoherence. Could the authors clarify this point?

Our response: The nitric acid soak is only used for the transient device.

Therefore, for lines 296-297, we are discussing the CD treated following the acetone soak. The metal composition is Ag and Au and we can use the Ag present in the CD to create a reference electrode.

- **Comment (6)** Lines 69-73: “A fully fabricated UCDE device consisted of two biopotential electrodes, a heater or temperature sensor, a reference electrode, a counter electrode, a pH electrode, an oxygen electrode, a lactate electrode, and a glucose electrode (Fig. 1K). The full, end-to-end fabrication and manufacturing required resources that can be found easily at conventional craft stores, negating the need for high-end instrumentation. “

The authors say that the fully UCDE is composed by 2 biopotential electrodes , but immediately describe that there are electrode to measure: pH, oxygen, lactate and glucose (4). Please clarify this point.

Our response: Yes, the full device consists of 2 biopotential electrodes, 6 electrochemical electrodes, and a heater/RTD. The configuration is presented in Figure 1J–L. This is the design of the device we engineered and proposed. Any configuration can be developed based on the user’s design. If researchers wanted a separate biopotential, electrochemical, or heater/RTD device, that can be easily created. For simplicity, we proposed a full device consisting of all these components, rather than 3 separate devices.

- **Comment (7)** Lines 178-180: “A Clark type oxygen sensor was based on the interaction of Nafion and a diluted PDMS layer (oxygen selective membrane) coating the UCDE’s electrode following electrochemical cleaning”. How is the oxygen sensor calibrated? Usually Clark electrode requires frequent calibration.

Our response: The oxygen sensor is calibrated against a commercial dissolved oxygen probe. The procedure is described in the supplementary section entitled, “Electrochemical Performance Evaluation”.

- **Comment (8)** Lines 196-201: “The UCDE’s glucose sensor produced a dynamic range between 0.15 mM to 0.75 mM at a sensitivity of - 0.94 $\mu\text{A}/\text{cm}^2\text{mM}$ ($R^2 = 0.98$) and limit of detection, 0.75 mM, with physiologically relevant concentrations for sweat glucose levels, 0.2 to 0.6 mM . The UCDE’s lactate sensor demonstrated a dynamic range from 3 to 9 mM with a sensitivity of -21.5 $\text{nA}/\text{cm}^2 \text{mM}$ ($R^2 = 0.98$) and limit of detection, 12 mM, ...”

Here there is a misunderstanding on the concept of “limit of detection “. According to IUPAC: The limit of detection, expressed as the concentration, c_L , or the quantity, q_L , is derived from the smallest measure, x_L , that can be detected with reasonable certainty for a given analytical procedure. The value of x_L is given by the equation $x_L = \bar{x}_b + k s_b$ where \bar{x}_b is the mean of the blank measures, s_b is the standard deviation of the blank measures, and k is a numerical factor chosen according to the confidence level desired. Probably the authors interpreted this values as the linear range or the dynamic range .

(IUPAC. Compendium of Chemical Terminology, 2nd ed. (the "Gold Book"). Compiled by A. D. McNaught and A. Wilkinson. Blackwell Scientific Publications, Oxford (1997). Online version (2019-) created by S. J. Chalk. ISBN 0-9678550-9-8. <https://doi.org/10.1351/goldbook.>).

Our response: We thank the reviewer for pointing out this typographical error. We completely agree that the term, “limit of detection” was misused. We meant to write limit of linearity and we made the changes in the manuscript.

Modifications to the manuscript: (Main Text, line 196–201) “The UCDE’s glucose sensor produced a dynamic range between 0.15 mM to 0.75 mM at a sensitivity of -0.94 $\mu\text{A}/\text{cm}^2\text{mM}$ ($R^2 = 0.98$) and limit of linearity, 0.75 mM, with physiologically relevant concentrations for sweat glucose levels, 0.2 to 0.6 mM^{56} . The UCDE’s lactate sensor demonstrated a dynamic range from 3 to 9 mM with a sensitivity of -21.5 $\text{nA}/\text{cm}^2 \text{mM}$ ($R^2 = 0.98$) and limit of linearity, 12 mM...”

- **Comment (9)** Another aspect not clear is how do the authors intend to do the quantification of the analyte, since it is not possible to calculate the concentration from a calibration plot obtained from standard additions of the analytes, as presented in the work. This is an tricky point common to almost all electrochemical sensors. The authors did not gave information about the reproducibility of the fabrication and well as of the measurements. It is necessary to have a statistical study on the performance of the sensors.

Our response: We have demonstrated similar procedures to Wei Gao et al. and cited them within the electrochemistry section describing our results [1]. We present a calibration curve that can be used to estimate the concentration. Enzymatic sensors also suffer from being unstable and typically suffer from a short shelf-life. We hope that our development here, as a prototyping method could help researchers address these types of issues to improve sensor reliability moving forward.

1. Gao, W., et al. Fully integrated wearable sensor arrays for multiplexed in situ perspiration analysis. *Nature* **529**, 509-514 (2016)

- **Comment (10)** Lines 253-255: “We hypothesize that *in vivo*, multinucleated macrophages, multinucleated giant cells, or foreign body giant cells would be able to clear out these flakes through phagocytosis at the expense of an elevated inflammatory response “. This hypothesis should be confirmed.

Our response: We agree, but this is a communications paper, and not the main focus of our study. We have presented previous literature that confirms our hypothesis [1]. In our future study, we will devote our efforts to the transient CD electronics to fully realize the potential of this device.

1. Carlander, U., et al. Macrophage-Assisted Dissolution of Gold Nanoparticles. *ACS Appl. Bio Mater.* **2**, 1006-1016 (2019).

We added an additional reference examining macrophage uptake of gold nanoparticles.

Modifications to the manuscript: (Main Text, lines 253–255) “We hypothesize that *in vivo*, multinucleated macrophages, multinucleated giant cells, or foreign body giant cells would be able to clear out these flakes through phagocytosis at the expense of an elevated inflammatory response^{66, 67}.”

(Reference update):

66. Xia, Z. & Triffitt, J. T. A review on macrophage responses to biomaterials. *Biomed. Mater.* **1**, R1-R9 (2006).
67. Carlander, U., et al. Macrophage-Assisted Dissolution of Gold Nanoparticles. *ACS Appl. Bio Mater.* **2**, 1006-1016 (2019).

- **Comment (11)** Lines 294-295: “Electrochemical Cleaning: All electrodes (except the reference and pH electrode) were cleaned in 0.1 M H2SO4 from -0.4 V to 1.4 V (vs. Ag/AgCl (1M KCl)) at 25 mV/s for 1 cycle.”

The cleaning of gold electrode surface is very tricky and important. Usually just one CV cycle is not enough to clean a gold electrode. See: Campos et al, <https://doi.org/10.1016/j.electacta.2013.07.083> for cleaning a gold electrode.

Our response: We agree, cleaning is an important and difficult step, usually 1 cleaning cycle is not enough. But, after cleaning with 1 cycle we can see a clean redox reaction within potassium ferrocyanide (Figure S15). We suspect that since the material is very thin and delicate, only 1 cycle is required to maintain the robustness of the whole device. It is also important to note that we are using a very slow scan rate of 25 mV/s. When we performed additional cycles, we noticed the performance of the electrode degraded substantially.

- **Comment (12)** Lines 296-297: “Reference Electrode: The reference electrode was fabricated by utilizing the trace amount of silver within the active electrode material from the CD.”

Here the authors indicate that the material of the CD useful as active electrode contains trace amount of silver. See a previous comment.

Probably many American researchers use hr and hrs to represent hour and hours. However the international abbreviation is h. I consider that it is important to respect international rules.

Our response: We addressed this concern with the previous comment. Please see above response. To respect international rules, we have changed hr. and hrs. within our manuscript.

- **Comment (13)** Lines 313-315: “The solution of Prussian Blue consisted of 100 mM KCl, 2.5 mM K₃Fe(CN)₆, 2.5 mM FeCl₃, and 100 mM HCl. For the lactate sensor, the Prussian Blue mediator layer was electrochemically deposited through cyclic voltammetry from -0.5 V to 0.6 V (vs. Ag/AgCl).”

It is well known that Prussian Blue is not soluble in water, so you cannot prepare a “solution of Prussian Blue”. What is indicated in this text is the solution present in the cell for the electrochemical deposition of PB. Please correct.

Our response: Correct, this is the Prussian Blue (PB) solution within the electrochemical cell. Here, we are electrochemically depositing the PB through cyclic voltammetry, which has been widely studied and utilized for decades. We modified the procedure described by Gao et al. for the CD electrodes. Through potentiodynamic and potentiostatic techniques a solution of PB can be deposited on working electrodes. This technique is widely used for oxidase enzymes to ensure great selectivity and dynamic range for the target analyte.

1. Gao, W, et al. Fully integrated wearable sensor arrays for multiplexed in situ perspiration analysis. Nature **529**, 509-514 (2016).

Reviewer #4

Summary Recommendation

The title of your manuscript is interesting given the importance of upcycling the waste resources within our technosphere. However, I find some bigger challenges not addressed by this manuscript.

Our response: We thank Reviewer 4 for their time and effort to review our work. We hope our revision addresses some of their main concerns.

- **Comment (1)** First of all, as you have mentioned yourself in line 36 that the compact discs are dated technologies, then why spend precious resources on figuring out a waste treatment option for them. An estimated 5.5. million CDs are discarded currently, but what about coming years when they wouldn't appear anymore in our waste flows. Already, we don't see them in noticeable number/amount in e-waste pre-processing facilities in many countries. In such a case, are you looking to dig up the urban mines (landfills) to get the CDs for your upcycling process? and how environment friendly and sustainable that's going to be?

Our response: We agree, CDs are dated technologies, however, they're still a large polluter within our waste stream, and they're still shipped at large quantities. In 2021, 40.6 million CDs were distributed in the US, a 1.1% increase from the previous year [1]. We couldn't track down any global numbers for this, however, we can assume that global shipments will be much higher. Additionally, this metric is only considering music CDs and excluding other types such as DVDs, video games, and software discs. This was surprising to us too! CDs are still being produced and shipped at larger values than you may think. Most of the e-waste seen today is from dated technologies, such as CDs [2].

We believe that Universities can install CD collection boxes, while Companies that provide CD collection methods, such as GreenDisk, may adopt or outsource the proposed fabrication techniques as an alternative to alleviate the accumulation of CDs in landfills.

We included a U.S. domestic policy that was recently approved for recycling. Although we would like to have a global perspective, we believe that since the U.S. is one of the leading producers of e-waste, these policies will have a bigger impact on the global scale.

Please see our detailed revision within the manuscript discussion possibilities.

1. MRC Data's 2021 U.S. Year-End Report. MRC Data (2022). https://mrcdatareports.com/wpcontent/uploads/2022/01/MRC_YEAREND_2021_US_FNL.pdf
2. A New Circular Vision for Electronics: Time for a Global Reboot. World Economic Forum (2019). https://www3.weforum.org/docs/WEF_A_New_Circular_Vision_for_Electronics.pdf

Modifications to our manuscript: (Introduction, line 29–53) “The disposal of electronic waste(e-waste) has become a concerning and growing waste stream driven by the short life cycle of electronics. In 2015, the United Nations established a blueprint for Sustainable Development Goals (SDGs)¹. The 12th SDG, “Responsible Consumption and Production”, seeks to address e-waste challenges by ensuring countries adopt a more responsible approach to the proliferating e-waste stream². Inefficient recycling processes are a global concern for e-waste management as they contribute to an increase in landfill waste and produce toxic pollution³. Additionally, Stephan Sicars (Director of the Department of Environment UN Industrial Development Organization) described e-waste as “a serious threat to the environment and human health worldwide”⁴. In 2019, the United Nations documented 1.7 kg per capita of e-waste recycled out of 7.3 kg per capita generated. To ensure the recycling of all e-waste by 2030 the recycling rate will need to be roughly 10 times greater². To reduce landfill and pollution accumulation, a more sustainable method is required to manage the flow of e-waste. Currently, only ~15–20% of e-waste is recycled despite its valuable materials—iron, steel, copper, silver, and gold^{5, 6, 7}. Whereas the remaining 80% of e-waste is not collected for recycling due to expense and lack of a global infrastructure^{5, 6, 7, 8}. Meanwhile, the toxic and hazardous components of e-waste—mercury, lead, and synthetic resins—threaten the environment and are left to degrade in landfills or incinerated^{5, 6, 7}. Today, e-waste primarily consists of dated technologies which accounts for the ever-growing trail⁵. Products from years past such as compact discs (CDs), old televisions, and computer monitors are the biggest contributors to e-waste⁵. Since 1999, 9.02 billion CDs have shipped in the United States⁹. In 2021, CD sales increased from the previous year by 1.1% to 40.6 million¹⁰. However, these statistics do not consider global shipments and only account for music CDs excluding other types such as DVDs, software discs, and video games. Thus, the global number of CDs produced and circulating globally are expected to be much larger. As societal dematerialization increases and we shift further towards digital streaming, where will all these CDs be deposited?”

(Reference update):

1. THE 17 GOALS. United Nations (2015). <https://sdgs.un.org/goals>
2. The Sustainable Development Goals Report. United Nations (2021). <https://unstats.un.org/sdgs/report/2021/The-Sustainable-Development-Goals-Report-2021.pdf>
3. Zhang, K., Schnoor, J. L. & Zeng, E. Y. E-Waste Recycling: Where Does It Go from Here? *Environ. Sci. Technol.* **46**, 10861-10867 (2012).
4. Environment and health at increasing risk from growing weight of ‘e-waste’. United Nations (2019). <https://news.un.org/en/story/2019/01/1031242>
5. A New Circular Vision for Electronics: Time for a Global Reboot. World Economic Forum (2019). https://www3.weforum.org/docs/WEF_A_New_Circular_Vision_for_Electronics.pdf
6. Wang, Z., Zhang, B. & Guan, D. Take responsibility for electronic-waste disposal. *Nature* **536**, 23-25 (2016).
7. Lee, B. & Chung, S. Printed carbon electronics get recycled. *Nat. Electron.* **4**, 241-242 (2021).

8. Forti, V., et al. *The Global E-waste Monitor 2020. Quantities, Flows, and the Circular Economy Potential* (2020).
9. Physical CD shipments in the United States from 1999 to 2020. Statista Research Department (2021). <https://www.statista.com/statistics/186772/album-shipments-in-the-us-music-industry-since-1999/>
10. MRC Data's 2021 U.S. Year-End Report. MRC Data (2022). https://mrcdatareports.com/wp-content/uploads/2022/01/MRC_YEAREND_2021_US_FNL.pdf

Modifications to our manuscript: (Conclusion, line 271–290) “Our paper illuminates the challenges that plague e-waste recycling and, subsequently, provides a remedy. Consumer confusion and a lack of infrastructure knowledge remain critical issues with regard to proper recycling. Emphasizing the development of novel upcycling approaches through scientific dissemination will increase awareness within this area. New programs incorporated into the Infrastructure Investment and Jobs Act, specifically the RECYCLE Act, aim to relieve the inundated recycling stream and provide new opportunities to support innovative recycling and upcycling ideas. Thus far, recycling and upcycling activities have accounted for 681,000 jobs, \$37.8 billion in wages, and \$5.5 billion in tax revenues in the United States⁶⁸. Policy programs that fund new jobs and ideas will assist the United States attain global sustainability objectives. Upcycling is a sustainable practice as it “meets the needs of the present without compromising the ability of future generations to meet their own needs”⁶⁹ through the transformation of waste into secondary products. The proposed upcycling approach remains sustainable if the cost of microfabrication remains exuberant, rapid prototyping persists as an essential business and institutional practice, and healthcare increasingly demands one-time use sensors. As a simple and cost-effective method, this technology can be adopted on both an academic research and commercial scale. Institutes and Universities may install CD collection boxes, while companies that provide CD collection methods, such as GreenDisk, may adopt or outsource the proposed fabrication techniques as an alternative to alleviate the accumulation of CDs in landfills. Any effort towards increasing both recycling and upcycling will advance the establishment of environmentally sustainable practices.”

(Reference update):

68. Recycling Economic Information. United States Environmental Protection Agency (2020). https://www.epa.gov/sites/default/files/2020-11/documents/rei_report_508_compliant.pdf
69. Our Common Future (Brundtland Report). Brussels: World Commission on Environment and Development (1987).

- **Comment (2)** Secondly, you mentioned in line 33 that 15-20% of e-waste gets recycled despite its valuable materials. Let me clarify for you that valuable materials are the first resources recovered from e-waste irrespective of developed or developing nations. So, this 15-20% could be a representative of valuable materials found in e-waste, where the recycling companies are simply not interested in recovering the remaining bulk for economic reasons. So, rather than focusing on obsolete technologies, it would be good to

devote all our energies to figure out ways for the remaining bulk materials, e.g., plastics and glass.

Our response: Some e-waste is recycled (approximately 15-20%). The valuable materials are recovered first, however, there is still 80% of e-waste that is unaccounted for. The 15-20% represents e-waste that is documented and recycled. Whereas the remaining 80% of e-waste is not collected for recycling due to expense and a lack of global infrastructure [1, 2, 3]. Our paper illuminates this challenge and provides a remedy.

See previous comment and associated revision explaining our rationale for CDs as a valuable e-waste stream.

1. Forti, V., et al. The Global E-waste Monitor 2020. Quantities, Flows, and the Circular Economy Potential (2020).
2. A New Circular Vision for Electronics: Time for a Global Reboot. World Economic Forum (2019).
https://www3.weforum.org/docs/WEF_A_New_Circular_Vision_for_Electronics.pdf
3. Wang, Z., Zhang, B. & Guan, D. Take responsibility for electronic-waste disposal. Nature **536**, 23-25 (2016).

- **Comment (3)** *It would be good to add a table of input and outputs for the upcycling process to visualize how many resources are used to little benefit. Also, what about a constructive discussion on demand for bioelectronic applications?*

Our response: We agree, and we have included a discussion within paragraph 2 of the introduction for the demand of bioelectronics.

We believe that a table of input and outputs may not be in our best interest for the research discussion of this manuscript. The only component not utilized through the upcycling process is the polycarbonate substrate which is separated from the metal layer after soaking in acetone. We would be happy to continue with follow up studies in the future and we may include this table in a review paper or prospectus article.

REVIEWER COMMENTS

Reviewer #1 (Remarks to the Author):

The authors have addressed most of the comments; they have also tried to make changes according to the reviewers' suggestions. After revisions, the quality of the manuscript has been adequately enhanced. Therefore, the manuscript could be considered for the publication.

Reviewer #3 (Remarks to the Author):

My comments to the reviewed version of the manuscript:

Upcycling Compact Discs for Flexible and Stretchable Bioelectronic Applications, Matthew S. Brown, Louis Somma, Melissa Mendoza, Yeonsik Noh, Gretchen J. Mahler, and Ahyeon Koh

Looking all the comments made by the reviewers, I observed the main comments of reviewers 1,3 and 4 are very similar, concerning to originality and real environmental effect of the recycling of CD and DVD. The observations were satisfactorily answered by the authors. In my opinion, in this final version of the manuscript a wide window is opened on economical and industrial aspects behind some problems faced in terms of e-waste. This is an interesting global issue!

In the response given by the authors to the question of considering CD as an old technology, the authors wrote in the first paragraph of the Introduction: "However, these statistics do not consider global shipments and only account for music CDs excluding other types such as DVDs, software discs, and video games." I would like to mention the increasing use of CD/DVD in the biomedical field to present the imaging and results of CT scan, magnetic resonance, conventional radiology, senology, angiography, etc. Other uses of CD/DVD can appear in the future, suggesting that the problem of this waste is not at the end.

I do not agree with the response given by the authors to one of my comments. It is not a fundamental question, but it is something wrong from chemical point of view.

- Comment (13) Lines 313-315: "The solution of Prussian Blue consisted of 100 mM KCl, 2.5 mM $K_3Fe(CN)_6$, 2.5 mM $FeCl_3$, and 100 mM HCl. For the lactate sensor, the Prussian Blue mediator layer was electrochemically deposited through cyclic voltammetry from -0.5 V to 0.6 V (vs. Ag/AgCl)."

It is well known that Prussian Blue is not soluble in water, so you cannot prepare an aqueous solution of Prussian Blue. What is indicated in this text is the solution present in the cell for the electrochemical deposition of PB. Please correct.

Our response: Correct, this is the Prussian Blue (PB) solution within the electrochemical cell. Here, we are electrochemically depositing the PB through cyclic voltammetry, which has been widely studied and utilized for decades. We modified the procedure described by Gao et al. for the CD electrodes. Through potentiodynamic and potentiostatic techniques a solution of PB can be deposited on working electrodes. This technique is widely used for oxidase enzymes to ensure great selectivity and dynamic range for the target analyte.

Any chemist could criticise this very basic chemistry. I know very well that PB is widely used as mediator in electrochemical sensors and biosensors and how it is prepared. The solution in the cell is not a PB solution, it is a solution of the reagents ($K_3Fe(CN)_6$ and $FeCl_3$) used to electrochemically (by cyclic voltammetry) produce and deposit a thin layer of PB on the electrode surface.

In the reference indicated by the authors (Gao, W, et al. Fully integrated wearable sensor arrays for multiplexed in situ perspiration analysis. Nature 529, 509-514 (2016).) it is correctly written: "A Prussian blue mediator layer was deposited onto the Au electrodes by cyclic voltammetry from 0V to 0.5 V (versus Ag/AgCl) for one cycle at a scan rate of 20 mVs⁻¹ in a fresh solution containing 2.5 mM $FeCl_3$, 100 mM KCl, 2.5 mM $K_3Fe(CN)_6$, and 100 mM HCl."

I hope that now my observation is clear.

Reviewer #5 (Remarks to the Author):

In this MS, a resource utilization method for directly preparing waste compact discs (CDs) into a wearable electronic device is proposed. This technology not only solves the pollution problem of toxic substances (Bisphenol A (BPA)) in waste CDs, but also finds a good source of raw materials for flexible materials. This technology is not a simple recycling of waste CDs, but proposes an advanced technological strategy from waste to high-performance emerging products. I am happy to see that this research can be published in Nature Communications to meet more readers after solving the following problems.

1. Abstract should provide more technical details and performance parameter information, rather than simply describe the technical effect, which is difficult to be convincing.
2. In line 10 of main text, how to dispose of acetone waste with polycarbonate substrates dissolved? This may be converted into a volatile organic waste.
3. Please supplement the ordinate units in Figure 1F.
4. Figure 5E of main text gives 400 seconds of stability test, is it enough to express its stability performance?
5. The conclusion part needs to briefly describe the current research level of waste CDs utilization, as well as the technical contribution of this research to this field and the upgrade of strategy or theory.

Modifications to the manuscript: (Data availability, line 374–380) “All relevant data supporting the fabrication, testing, and functionalization of the sensors within the study are presented in the paper and Supplementary Information file. Additional information may be requested from the corresponding author upon reasonable request. Further details are described within the Supplementary Information file that includes an alternative fabrication method, temperature sensor calibration, electrochemical performance evaluation, the fabrication of the biodegradable UCDEs, and solutions to recycle acetone and polycarbonate.”

Reviewer #1

Summary Recommendation

The authors have addressed most of the comments; they have also tried to make changes according to the reviewers' suggestions. After revisions, the quality of the manuscript has been adequately enhanced. Therefore, the manuscript could be considered for the publication.

Our response: We thank Reviewer 1 for their time and effort to review our work. We are pleased they feel our revisions have improved the quality of the manuscript.

Reviewer #3

Summary Recommendation

Looking all the comments made by the reviewers, I observed the main comments of reviewers 1,3 and 4 are very similar, concerning to originality and real environmental effect of the recycling of CD and DVD. The observations were satisfactorily answered by the authors. In my opinion, in this final version of the manuscript a wide window is opened on economical and industrial aspects behind some problems faced in terms of e-waste. This is an interesting global issue!

In the response given by the authors to the question of considering CD as an old technology, the authors wrote in the first paragraph of the Introduction: “However, these statistics do not consider global shipments and only account for music CDs excluding other types such as DVDs, software discs, and video games.” I would like to mention the increasing use of CD/DVD in the biomedical field to present the imaging and results of CT scan, magnetic resonance, conventional radiology, senology, angiography, etc. Other uses of CD/DVD can appear in the future, suggesting that the problem of this waste is not at the end.

Our response: We thank Reviewer 3 for reconsidering our work.

Modifications to our manuscript: (Introduction, line 59–64) “Furthermore, the biomedical field utilizes CDs as a primary medium for medical images between both patients and providers. Thus, the global number of CDs produced and circulating globally are expected to be much larger and the end of the CD waste stream remains unclear. As societal dematerialization increases and we shift further towards electronic platforms, where will all these CDs be deposited?”

- *Comment (1) I do not agree with the response given by the authors to one of my comments. It is not a fundamental question, but it is something wrong from chemical point of view.*

• Comment (13) Lines 313-315: “The solution of Prussian Blue consisted of 100 mM KCl, 2.5 mM K₃Fe(CN)₆, 2.5 mM FeCl₃, and 100 mM HCl. For the lactate sensor, the Prussian Blue mediator layer was electrochemically deposited through cyclic voltammetry from -0.5 V to 0.6 V (vs. Ag/AgCl).”

It is well known that Prussian Blue is not soluble in water, so you cannot prepare an aqueous solution of Prussian Blue. What is indicated in this text is the solution present in the cell for the electrochemical deposition of PB. Please correct.

Our response: Correct, this is the Prussian Blue (PB) solution within the electrochemical cell.

Here, we are electrochemically depositing the PB through cyclic voltammetry, which has been widely studied and utilized for decades. We modified the procedure described by Gao et al. for the CD electrodes. Through potentiodynamic and potentiostatic techniques a solution of PB can be deposited on working electrodes. This technique is widely used for oxidase enzymes to ensure great selectivity and dynamic range for the target analyte.

Any chemist could criticise this very basic chemistry. I know very well that PB is widely used as mediator in electrochemical sensors and biosensors and how it is prepared. The solution in the cell is not a PB solution, it is a solution of the reagents (K₃Fe(CN)₆ and FeCl₃) used to electrochemically (by cyclic voltammetry) produce and deposit a thin layer of PB on the electrode surface.

In the reference indicated by the authors (Gao, W, et al. Fully integrated wearable sensor arrays for multiplexed in situ perspiration analysis. Nature 529, 509-514 (2016).) it is correctly written:

“A Prussian blue mediator layer was deposited onto the Au electrodes by cyclic voltammetry from 0 V to 0.5 V (versus Ag/AgCl) for one cycle at a scan rate of 20 mV s⁻¹ in a fresh solution containing 2.5 mM FeCl₃, 100 mM KCl, 2.5 mM K₃Fe(CN)₆, and 100 mM HCl.”

I hope that now my observation is clear.

Our response: We apologize for the misinterpretation on our end. We thank you for bringing up this point and we acknowledge that we used incorrect wording.

Modifications to our manuscript: (Methods, line 349–350) “The Prussian Blue mediator layer was electrochemically deposited in a fresh solution of 100 mM KCl, 2.5 mM $K_3Fe(CN)_6$, 2.5 mM $FeCl_3$, and 100 mM HCl.”

Reviewer #4 and Editorial Comments

Summary Recommendation

It would be good to add a table of input and outputs for the upcycling process to visualize how many resources are used to little benefit.

Our response: We added a flow chart to the supplementary section, and we hope that it will enhance our paper.

Modifications to our manuscript: (Main Text, line 10–11) “The inputs and outputs of the upcycling fabrication process vs. microfabrication are illustrated in Supplementary Figure 4”

(Supplementary, Fig. 4)

Supplementary Figure 4. Inputs and outputs of the upcycling process vs. microfabrication.

Reviewer #5

Summary Recommendation

In this MS, a resource utilization method for directly preparing waste compact discs (CDs) into a wearable electronic device is proposed. This technology not only solves the pollution problem of toxic substances (Bisphenol A (BPA)) in waste CDs, but also finds a good source of raw materials for flexible materials. This technology is not a simple recycling of waste CDs, but proposes an advanced technological strategy from waste to high-performance emerging products. I am happy to see that this research can be published in Nature Communications to meet more readers after solving the following problems.

Our response: We appreciate the positive comments from Reviewer 5.

- *Comment (1) Abstract should provide more technical details and performance parameter information, rather than simply describe the technical effect, which is difficult to be convincing.*

Modifications to our manuscript: (Abstract, line 24–32) “Upcycled CD electronics (UCDEs) as stretchable electromyography and electrocardiogram electrodes performed analogous to commercial gel electrodes, presenting a comparable signal-to-noise ratio. At 5 V, the wearable UCDE heater emitted an average heat output of 35.6 °C. The resistance temperature detector presented similar sensitivity to a laboratory based infrared camera. Potentiometric, amperometric, and enzymatic-based UCDEs were demonstrated as pH (4–12 pH), oxygen (20.2–100 O₂%), glucose (0.15–0.75 mM), and lactate (3–9 mM) sensors exhibiting physiologically relevant performance. Lastly, UCDEs as transient resistors were fabricated, enabling biodegradable electronics for fully recyclable devices”

- *Comment (2) In line 10 of main text, how to dispose of acetone waste with polycarbonate substrates dissolved? This may be converted into a volatile organic waste.*

Our response: The polycarbonate substrate is not fully dissolved within the acetone solution (see Supplementary Figure 7C). Since the CD is only soaked for 1.5 minutes, the concentration is very low (<μg). Thus, the acetone can be recovered through distillation which is commonly used commercially to remove contaminants from the solvent. Many companies sell these solvent recyclers. The acetone can also be outsourced to recycling companies that have this in-house recycling equipment.

We used FTIR to determine if polycarbonate can be detected within the acetone solvent following the soaking of the CD. We added this figure to the supplementary section.

We added to the supplementary section a solution on how to recycle the acetone waste and polycarbonate substrate.

(Supplementary, Fig. 6B)

Modifications to our manuscript: (Main Text, line 13–14): “However, the concentration within the acetone was undetectable (Supplementary Fig. 6B).”

(Supplementary Information, line 85–108): The UCDEs were upcycled with acetone to break down the polycarbonate and allow for the simple release of the metal layer. The minimum volume required is 40 mL and can be reused with multiple CDs. The procedure requires the CDs soaked for a short period of time, thus, the polycarbonate dissolved within acetone is less than μg concentrations and even soaking the CD for 1 hour, no large peak changes can be detected by FTIR (Supplementary Fig. 6B). After the metal layer is stripped from the CD, the polycarbonate substrate remains. The CD can no longer be classified as laced e-waste (mixed plastic and metal) and can be recycled through conventional methods. Here we discuss solutions to recycle the acetone and polycarbonate substrate.

For decades acetone has been used by many industries to clean large scale and laboratory equipment. When acetone is used and contaminated it is shipped to off-site locations for proper disposal. Recovery and recycling of acetone can be utilized to alleviate the cost and environmental impact of disposing acetone. Through simple distillation, this process can be operated through a continuous or batchwise process³. Solvent recyclers (e.g., NexGen Enviro Systems, Inc.) are commercially available, allowing for high-performance removal of contaminants and recovery of the solvent. However, these systems are quite expensive. To alleviate these costs, Zweckmair et al. details a simple automated process to recycle acetone through a home-built distillation unit⁴.

Many plastic recycling companies process polycarbonate. The polycarbonate is processed accordingly: (1) shredded and grinded into granules, (2) washed to clean contaminants, (3) densified, (4) blended and reprocessed into a resin or pellet, (5) compounded with additives. Researchers have also been looking for solutions to address this issue at the lab scale. Jones et al. recently developed a method to synthesis poly(aryl ether sulfone)s (PSUs) from the depolymerization of polycarbonate CDs⁵. The high performance engineered thermoplastics, PSUs can be repurposed for medical equipment or water purification applications⁵.

(Reference updates):

3. Drueckhammer, D. G. Acetone as a Recyclable Solvent. In: *Acetone: Biochemistry, Production and Uses* (ed Irenka Kozłowska, K. W.) 187-199 (Nova Science Publishers, 2018).
4. Zweckmair, T., *et al.* Recycling of Analytical Grade Solvents on a Lab Scale with a Purpose-Built Temperature-Controlled Distillation Unit. *Organic Process Research & Development* **21**, 578-584 (2017).
5. Jones, G. O., *et al.* Computational and experimental investigations of one-step conversion of poly(carbonate)s into value-added poly(aryl ether sulfone)s. **113**, 7722-7726 (2016).

- *Comment (3) Please supplement the ordinate units in Figure 1F.*

Modifications to our manuscript: (Figure 1F): In the y-axis we added “(arb. units)” and put the x-axis units in parentheses to remain consistent with the rest of the manuscript.

- *Comment (4) Figure 5E of main text gives 400 seconds of stability test, is it enough to express its stability performance?*

Our response: We presented the long-term stability within PBS in Supplementary Figure 24. This part of the manuscript is purely a proof-of-concept, and we don't regard it as the main significance of our proposed study. In the future we will perform more long-term studies to evaluate the stability long-term. We added a sentence within the conclusion discussing the future work to address this.

- *Comment (5) The conclusion part needs to briefly describe the current research level of waste CDs utilization, as well as the technical contribution of this research to this field and the upgrade of strategy or theory.*

Modifications to our manuscript: (Conclusion, line 258–286) “To date, researchers have presented methods to recycle CD waste into electrochemical sensors for scalable and cheap protocols. However, thus far, they have failed to demonstrate mechanically durable biosensor platforms for practical wearable applications. Our study addresses this limitation. These CDs can be transformed into soft bioelectronics for noninvasive monitoring, while fully integrating with

human skin. A mechanical machine cutter carefully defined the UCDEs for affordable micropatterning of fully stretchable and flexible electronics. We introduce a new upcycling approach and applications for fully scalable biopotential (EMG and ECG) sensing, heat emittance, temperature detection, electrochemical monitoring (pH, oxygen, lactate, and glucose), and moisture triggered transient sensing. Biopotential sensors developed showed similar performance to commercially available gel electrodes. The heater demonstrated produced an average heat output of 35.6 °C at 5 V and the RTD sensor exhibited analogous sensitivity to a laboratory based infrared camera. Potentiometric pH sensing illustrated a dynamic range from 4–12 pH and a sensitivity of -36.5 mV/decade. The amperometric sensor performance for the oxygen sensor was 20.2–100 O₂% (sensitivity = -65 nA/(cm²O₂%), glucose sensor was 0.15–0.75 mM (sensitivity = 0.94 μA/cm²mM), and lactate sensor was 3–9 mM (sensitivity = -21.5 nA/cm² mM). Lastly, transient performance was demonstrated for fully recyclable electronics. This translational development was fully optimized, to produce biologically relevant results in stretchability and flexibility as well as sensing performance while remaining fully biocompatible. Overall, this study provides a useful alternative for e-waste management, one-time use electronics, rapid prototyping, and inexpensive approaches for bioelectronic fabrication methods.

The proposed upcycling process reported here enables sustainable solutions for CDs and other e-waste recycling which can be demonstrated in the future. We foresee additional work towards: (1) evaluating the long-term performance of the electrochemical sensors, (2) fully integrated systems for wireless monitoring, (3) additional studies to fully realize the potential of the transient devices in applications towards implantable bioelectronics. The upcycling method provided here will allow for bioelectronic fabrication without the need for intensive training and microfabrication techniques, which opens the door to a wider variety of disciplines adopting stretchable and flexible based devices for their studies.”

REVIEWERS' COMMENTS

Reviewer #5 (Remarks to the Author):

The author has solved our questions and we recognize that this study is worth publishing.